# Boundary thickness and robustness
# in learning models

**Yaoqing Yang, Rajiv Khanna, Yaodong Yu, Amir Gholami,**
**Kurt Keutzer, Joseph E. Gonzalez, Kannan Ramchandran, Michael W. Mahoney**
University of California, Berkeley
Berkeley, CA 94720
`{yqyang, rajivak, yaodong_yu, amirgh, keutzer, jegonzal, kannanr,`
`mahoneymw}@berkeley.edu`

## Abstract

Robustness of machine learning models to various adversarial and non-adversarial corruptions continues to be of interest. In this paper, we introduce the notion of the boundary thickness of a classifier, and we describe its connection with and usefulness for model robustness. Thick decision boundaries lead to improved performance, while thin decision boundaries lead to overfitting (e.g., measured by the robust generalization gap between training and testing) and lower robustness. We show that a thicker boundary helps improve robustness against adversarial examples (e.g., improving the robust test accuracy of adversarial training) as well as so-called out-of-distribution (OOD) transforms, and we show that many commonly-used regularization and data augmentation procedures can increase boundary thickness. On the theoretical side, we establish that maximizing boundary thickness during training is akin to the so-called mixup training. Using these observations, we show that noise-augmentation on mixup training further increases boundary thickness, thereby combating vulnerability to various forms of adversarial attacks and OOD transforms. We can also show that the performance improvement in several lines of recent work happens in conjunction with a thicker boundary.

## 1 Introduction

Recent work has re-highlighted the importance of various forms of robustness of machine learning models. For example, it is by now well known that by modifying natural images with barely-visible perturbations, one can get neural networks to misclassify images [1–4]. Researchers have come to call these slightly-but-adversarially perturbed images *adversarial examples*. As another example, it has become well-known that, even aside from such worst-case adversarial examples, neural networks are also vulnerable to so-called *out-of-distribution (OOD) transforms* [5], i.e., those which contain common corruptions and perturbations that are frequently encountered in natural images. These topics have received interest because they provide visually-compelling examples that expose an inherent lack of stability/robustness in these already hard-to-interpret models [6–12], but of course similar concerns arise in other less visually-compelling situations.

In this paper, we study neural network robustness through the lens of what we will call *boundary thickness*, a new and intuitive concept that we introduce. Boundary thickness can be considered a generalization of the standard margin, used in max-margin type learning [13–15]. Intuitively speaking, the boundary thickness of a classifier measures the expected distance to travel along line segments between different classes across a decision boundary. We show that thick decision boundaries have a regularization effect that improves robustness, while thin decision boundaries lead to overfitting and reduced robustness. We also illustrate that the performance improvement in several lines of

recent work happens in conjunction with a thicker boundary, suggesting the utility of this notion more generally.

More specifically, for adversarial robustness, we show that five commonly used ways to improve robustness can increase boundary thickness and reduce the robust generalization gap (which is the difference between robust training accuracy and robust test accuracy) during adversarial training. We also show that trained networks with thick decision boundaries tend to be more robust against OOD transforms. We focus on *mixup training* [16], a recently-described regularization technique that involves training on data that have been augmented with pseudo-data points that are convex combinations of the true data points. We show that mixup improves robustness to OOD transforms, while at the same time achieving a thicker decision boundary. In fact, the boundary thickness can be understood as a dual concept to the mixup training objective, in the sense that the former is maximized as a result of minimizing the mixup loss. In contrast to measures like margin, boundary thickness is easy to measure, and (as we observe through counter examples) boundary thickness can differentiate neural networks of different robust generalization gap, while margin cannot.

For those interested primarily in training, our observations also lead to novel training procedures. Specifically, we design and study a novel noise-augmented extension of mixup, referred to as *noisy mixup*, which augments the data through a mixup with random noise, to improve robustness to image imperfections. We show that noisy mixup thickens the boundary, and thus it significantly improves robustness, including black/white-box adversarial attacks, as well as OOD transforms.

In more detail, here is a summary of our main contributions.

- We introduce the concept of boundary thickness (Section 2), and we illustrate its connection to various existing concepts, including showing that as a special case it reduces to margin.

- We demonstrate empirically that a thin decision boundary leads to poor adversarial robustness as well as poor OOD robustness (Section 3), and we evaluate the effect of model adjustments that affect boundary thickness. In particular, we show that five commonly used regularization and data augmentation schemes ($\ell_1$ regularization, $\ell_2$ regularization, large learning rate [17], early stopping, and cutout [18]) all increase boundary thickness and reduce overfitting of adversarially trained models (measured by the robust accuracy gap between training and testing). We also show that boundary thickness outperforms margin as a metric in measuring the robust generalization gap.

- We show that our new insights on boundary thickness make way for the design of new robust training schemes (Section 4). In particular, we designed a noise-augmentation training scheme that we call noisy mixup to increase boundary thickness and improve the robust test accuracy of mixup for both adversarial examples and OOD transforms. We also show that mixup achieves the minimax decision boundary thickness, providing a theoretical justification for both mixup and noisy mixup.

Overall, our main conclusion is the following.

> *Boundary thickness is a reliable and easy-to-measure metric that is associated with model robustness, and training a neural network while ensuring a thick boundary can improve robustness in various ways that have received attention recently.*

In order that our results can be reproduced and extended, we have open-sourced our code.[1]

**Related work.** Both adversarial robustness [1–4, 6–8, 11] and OOD robustness [5, 9, 19–21] have been well-studied in the literature. From a geometric perspective, one expects robustness of a machine learning model to relate to its decision boundary. In [1], the authors claim that adversarial examples arise from the linear nature of neural networks, hinting at the relationship between decision boundary and robustness. In [22], the authors provide the different explanation that the decision boundary is not necessarily linear, but it tends to lie close to the "data sub-manifold." This explanation is supported by the idea that cross-entropy loss leads to poor margins [23]. Some other works also study the connection between geometric properties of a decision boundary and the robustness of the model, e.g., on the boundary curvature [24, 25]. The paper [26] uses early-stopped PGD to

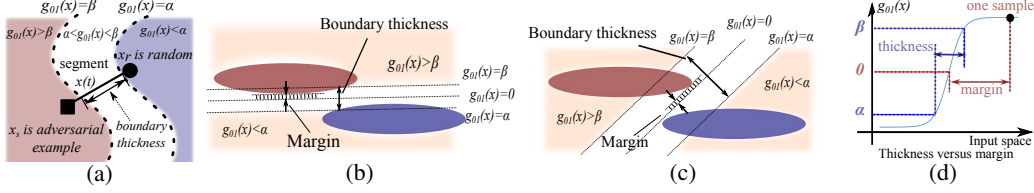

Figure 1: **Main intuition behind boundary thickness.** (a) Boundary thickness measures the gap between two level sets $g_{01}(x) = \alpha$ and $g_{01}(x) = \beta$ along the adversarial direction. (b) A thin boundary easily fits into the narrow space between two different classes, but it is not robust. (c) A thick boundary is harder to achieve a small loss, but it achieves higher robustness. In these two toy settings, the margin is the same, but choosing a thicker boundary leads to better robustness. (d) Illustration of the relationship between boundary thickness and margin.

improve natural accuracy, and it discusses how robust loss changes the decision boundary. Another related line of recent work points out that the inductive bias of neural networks towards "simple functions" may have a negative effect on network robustness [27], although being useful to explain generalization [28, 29]. These papers support our observation that natural training tends to generate simple, thin, but easy-to-attack decision boundaries, while avoiding thicker more robust boundaries.

Mixup is a regularization technique introduced by [16]. In mixup, each sample is obtained from two samples $x_1$ and $x_2$ using a convex combination $x = tx_1 + (1 - t)x_2$, with some random $t \in [0, 1]$. The label $y$ is similarly obtained by a convex combination. Mixup and some other data augmentation schemes have been successfully applied to improve robustness. For instance, [30] uses mixup to interpolate adversarial examples to improve adversarial robustness. The authors in [31] combine mixup and label smoothing on adversarial examples to reduce the variance of feature representation and improve robust generalization. [32] uses interpolation between Gaussian noise and Cutout [18] to improve OOD robustness. The authors in [9] mix images augmented with various forms of transformations with a smoothness training objective to improve OOD robustness. Compared to these prior works, the extended mixup with simple noise-image augmentation studied in our paper is motivated from the perspective of decision boundaries; and it provides a concrete explanation for the performance improvement, as regularization is introduced by a thicker boundary. Another recent paper [33] also shows the importance of reducing overfitting in adversarial training, e.g., using early stopping, which we also demonstrate as one way to increase boundary thickness.

## 2 Boundary Thickness

In this section, we introduce boundary thickness and discuss its connection with related notions.

### 2.1 Boundary thickness

Consider a classification problem with $C$ classes on the domain space of data $\mathcal{X}$. Let $f(x) : \mathcal{X} \to [0, 1]^C$ be the prediction function, so that $f(x)_i$ for class $i \in [C]$ represents the posterior probability $\Pr(y = i|x)$, where $(x, y)$ represents a feature vector and response pair. Clearly, $\sum_{i=0}^{C-1} f(x)_i = 1, \forall x \in \mathcal{X}$. For neural networks, the function $f(x)$ is the output of the softmax layer. In the following definition, we quantify the thickness of a decision boundary by measuring the posterior probability difference $g_{ij}(x) = f(x)_i - f(x)_j$ on line segments connecting pairs of points $(x_r, x_s) \in \mathcal{X}$ (where $x_r, x_s$ are *not* restricted to the training set).

**Definition 1** (Boundary Thickness). *For $\alpha, \beta \in (-1, 1)$ and a distribution $p$ over pairs of points $(x_r, x_s) \sim p$, let the predicted labels of $x_r$ and $x_s$ be $i$ and $j$ respectively. Then, the boundary thickness of a prediction function $f(\cdot)$ is*

$$\Theta(f, \alpha, \beta, p) = \mathbb{E}_{(x_r, x_s) \sim p} \left[ \|x_r - x_s\| \int_0^1 \mathbf{I}\{\alpha < g_{ij}(x(t)) < \beta\} dt \right], \qquad (1)$$

*where $g_{ij}(x) = f(x)_i - f(x)_j$, $\mathbf{I}\{\cdot\}$ is the indicator function, and $x(t) = tx_r + (1 - t)x_s, t \in [0, 1]$.*

Intuitively, boundary thickness captures the distance between two level sets $g_{ij}(x) = \alpha$ and $g_{ij}(x) = \beta$ by measuring the expected gap on random line segments in $\mathcal{X}$. See Figure 1a. Note that in addition

to the two constants, $\alpha$ and $\beta$, Definition 1 of boundary thickness requires one to specify a distribution $p$ to choose pairs of points $(x_r, x_s)$. We show that specific instances of $p$ recovers margin (Section 2.2) and mixup regularization (Section A). For the rest of the paper, we set $p$ as follows. Choose $x_r$ uniformly at random from the training set. Denote $i$ its predicted label. Then, choose $x_s$ to be an $\ell_2$ adversarial example generated by attacking $x_r$ to a random target class $j \neq i$. We first look at a simple example on linear classifiers to illustrate the concept.

**Example 1** (Binary Linear Classifier). Consider a binary linear classifier, with weights $w$ and bias $b$. The prediction score vector is $f(x) = [f(x)_0, f(x)_1] = [\sigma(w^\top x + b), 1 - \sigma(w^\top x + b)]$, where $\sigma(\cdot) : \mathbb{R} \to [0, 1]$ is the sigmoid function. In this case, measuring thickness in the $\ell_2$ adversarial direction means that we choose $x_r$ and $x_s$ such that $x_s - x_r = cw, c \in \mathbb{R}$. In the following proposition, we quantify the boundary thickness for a binary linear classifier. (See Section B.1 for the proof.)

**Proposition 2.1** (Boundary Thickness of Binary Linear Classifier). *Let $\tilde{g}(\cdot) := 2\sigma(\cdot) - 1$. If $[\alpha, \beta] \subset [g_{01}(x_r), g_{01}(x_s)]$, then the thickness of the binary linear classifier is given by:*

$$\Theta(f, \alpha, \beta) = (\tilde{g}^{-1}(\beta) - \tilde{g}^{-1}(\alpha))/\|w\|. \tag{2}$$

Note that $x_s$ and $x_r$ should be chosen such that the condition $[\alpha, \beta] \subset [g_{01}(x_r), g_{01}(x_s)]$ in Proposition 2.1 is satisfied. Otherwise, for linear classifiers, the segment between $x_r$ and $x_s$ is not long enough to span the gap between the two level sets $g_{01}(x) = \alpha$ and $g_{01}(x) = \beta$ and cannot simultaneously intersect the two level sets.

For a pictorial illustration of why boundary thickness should be related to robustness and why a thicker boundary should be desirable, see Figures 1b and 1c. The three curves in each figure represent the three level sets $g_{01}(x) = \alpha$, $g_{01}(x) = 0$, and $g_{01}(x) = \beta$. The thinner boundary in Figure 1b easily fits the narrow space between two different classes, but it is easier to attack. The thicker boundary in Figure 1c, however, is harder to fit data with a small loss, but it is also more robust and harder to attack. To further justify the intuition, we provide an additional example in Section C. Note that the intuition discussed here is reminiscent of max margin optimization, but it is in fact more general. In Section, 2.2, we highlight differences between the two concepts (and later, in Section 3.3, we also show that margin is not a particularly good indicator of robust performance).

## 2.2 Boundary thickness generalizes margin

We first show that boundary thickness reduces to margin in the special case of binary linear SVM. We then extend this result to general classifiers.

**Example 2** (Support Vector Machines). As an application of Proposition 2.1, we can compute the boundary thickness of a binary SVM, which we show is equal to the margin. Suppose we choose $\alpha$ and $\beta$ to be the values of $\tilde{g}(u)$ evaluated at two support vectors, i.e., at $u = w^\top x + b = -1$ and $u = w^\top x + b = 1$. Then, $\tilde{g}^{-1}(\alpha) = -1$ and $\tilde{g}^{-1}(\beta) = 1$. Thus, from (2), we obtain $(\tilde{g}^{-1}(\beta) - \tilde{g}^{-1}(\alpha))/\|w\| = 2/\|w\|$, which is the (input-space) margin of an SVM.

We can also show that the reduction to margin applies to more general classifiers. Let $S(i, j) = \{x \in \mathcal{X} : f(x)_i = f(x)_j\}$ denote the decision boundary between classes $i$ and $j$. The (input-space) margin [13] of $f$ on a dataset $\mathcal{D} = \{x_k\}_{k=0}^{n-1}$ is defined as

$$Margin(\mathcal{D}, f) = \min_k \min_{j \neq y_k} \|x_k - \text{Proj}(x_k, j)\|, \tag{3}$$

where $\text{Proj}(x, j) = \arg\min_{x' \in S(i_x, j)} \|x' - x\|$ is the projection onto the decision boundary $S(i_x, j)$. See Figure 1b and 1c. Boundary thickness for the case when $\alpha = 0$, $\beta = 1$, and when $p$ is so chosen that $x_s$ is the projection $\text{Proj}(x_r, j)$ for the worst case class $j$, reduces to margin. See Figure 1d for an illustration of this relationship for a two-class problem. Note that the left hand side of (4) is a "worst-case" version of the boundary thickness in (1). This can be formalized in the following proposition. (See Section B.2 for the proof.)

**Proposition 2.2** (Margin is a Special Case of Boundary Thickness). *Choose $x_r$ as an arbitrary point in the dataset $\mathcal{D} = \{x_k\}_{k=0}^{n-1}$, with predicted label $i = \arg\max_l f(x)_l$. For another class $j \neq i$, choose $x_s = Proj(x_r, j)$. Then,*

$$\min_{x_r} \min_{j \neq i} \|x_r - x_s\| \int_0^1 \mathbf{I}\{\alpha < g_{ij}(x(t)) < \beta\} dt = Margin(\mathcal{D}, f). \tag{4}$$

**Remark 1** (Margin versus Thickness as a Metric). It is often impractical to compute the margin for general nonlinear functions. On the other hand, as we illustrate below, measuring boundary thickness is straightforward. As noted by [16], using mixup tends to make a decision boundary more "linear," which helps to reduce unnecessary oscillations in the boundary. As we show in Section 4.1, mixup effectively makes the boundary thicker. This effect is not directly achievable by increasing margin.

## 2.3 A thick boundary mitigates boundary tilting

Boundary tilting was introduced by [22] to capture the idea that for many neural networks the decision boundary "tilts" away from the max-margin solution and instead leans towards a "data sub-manifold," which then makes the model less robust. Define the cosine similarity between two vectors $a$ and $b$ as:

$$\text{Cosine Similarity}(a,b) = |a^\top b| / (\|a\|_2 \cdot \|b\|_2). \tag{5}$$

For a dataset $D = \{(x_i, y_i)\}_{i \in I}$, in which the two classes $\{x_i \in D : y_i = 1\}$ and $\{x_i \in D : y_i = -1\}$ are linearly separable, boundary tilting can be defined as the worse-case cosine similarity between a classifier $w$ and the hard-SVM solution $w^* := \arg\min \|v\|_2$ s.t. $y_i v^\top x_i \geq 1, \forall i \in I$:

$$T(u) := \min_{v \text{ s.t. } \|v\|_2 = u \text{ and } y_i v^\top x_i \geq 1, \forall i} \text{Cosine Similarity}(v, w^*), \tag{6}$$

which is a function of the $u$ in the $\ell_2$ constraint $\|v\|_2 = u$. In the following proposition, we formalize the idea that boundary thickness tends to mitigate tilting. (See Section B.3 for the proof; and see also Figure 1b.)

**Proposition 2.3** (A Thick Boundary Mitigates Boundary Tilting). *The worst-case boundary tilting* $T(u)$ *is a non-increasing function of* $u$.

A smaller cosine similarity between the $w$ and the SVM solution $w^*$ corresponds to more tilting. In other words, $T(u)$ achieves the maximum when there is no tilting. From Proposition 2.1, for a linear classifier, we know that $u = \|w\|_2$ is inversely proportional to thickness. Thus, $T(u)$ is a non-decreasing function in thickness. That is, a thicker boundary leads to a larger $T(u)$, which means the worst-case boundary tilting is mitigated. We demonstrate Proposition 2.3 on the more general nonlinear classifiers in Section D.

# 3 Boundary Thickness and Robustness

In this section, we measure the change in boundary thickness by slightly altering the training algorithm in various ways, and we illustrate the corresponding change in robust accuracy. We show that across many different training schemes, boundary thickness corresponds strongly with model robustness. We observe this correspondence for both non-adversarial as well as adversarial training. We also present a use case illustrating why using boundary thickness rather than margin as a metric for robustness is useful. More specifically, we show that a thicker boundary reduces overfitting in adversarial training, while margin is unable to differentiate different levels of overfitting.

## 3.1 Non-adversarial training

Here, we compare the boundary thicknesses and robustness of models trained with three different schemes on CIFAR10 [34], namely training without weight decay, training with standard weight decay, and mixup training [16]. Note that these three training schemes impose increasingly stronger regularization. We use different neural networks, including ResNets [35], VGGs [36], and DenseNet [37].[2] All models are trained with the same initial learning rate of 0.1. At both epoch 100 and 150, we reduce the current learning rate by a factor of 10. The thickness of the decision boundary is measured as described in Section 2 with $\alpha = 0$ and $\beta = 0.75$. When measuring thickness on the adversarial direction, we use an $\ell_2$ PGD-20 attack with size 1.0 and step size 0.2. We report the average thickness obtained by repeated runs on 320 random samples, i.e., 320 random samples with their adversarial examples. The results are shown in Figure 2a.

From Figure 2a, we see that the thickness of mixup is larger than that of training with standard weight decay, which is in turn larger than that of training without weight decay. From the thickness drop at

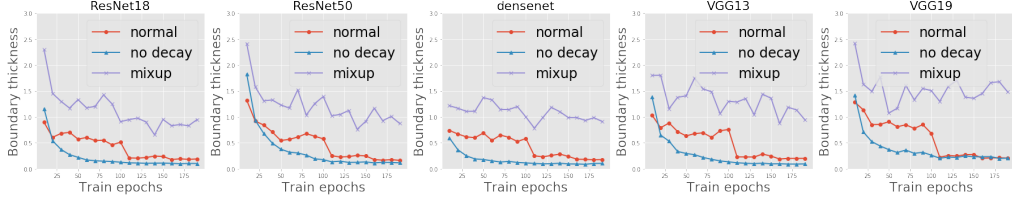

(a) Thickness: mixup > normal training > training without weight decay. After learning rate decays (at both epoch 100 and 150), decision boundaries get thinner.

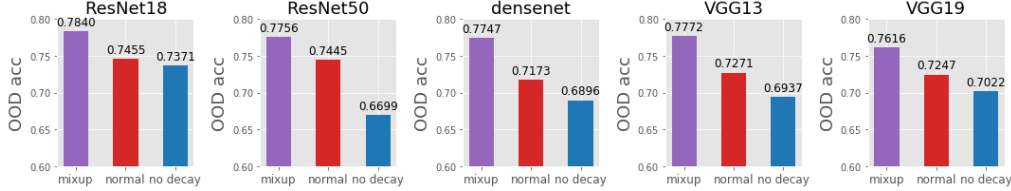

(b) OOD robustness: mixup > normal training > training without weight decay. Compare with Figure 2a to see that mixup increases thickness, while training without weight decay reduces thickness.

Figure 2: **OOD robustness and thickness.** OOD robustness improves with increasing boundary thickness for a variety of neural networks trained on CIFAR10. "Normal" means training with standard weight decay 5e-4, and "no decay" means training without weight decay. Mixup uses the recommended weight decay 1e-4.

epochs 100 and 150, we conclude that learning rate decay reduces the boundary thickness. Then, we compare the OOD robustness for the three training procedures on the same set of trained networks from the last epoch. For OOD transforms, we follow the setup in [38], and we evaluate the trained neural networks on CIFAR10-C, which contains 15 different types of corruptions, including noise, blur, weather, and digital corruption. From Figure 2b, we see that the OOD robustness corresponds to boundary thickness across different training schemes for all the tested networks.

See Section E.1 for more details on the experiment. See Section E.2 for a discussion of why the adversarial direction is preferred in measuring thickness. See Section E.3 for a thorough ablation study of the hyper-parameters, such as $\alpha$ and $\beta$, and on the results of two other datasets, namely CIFAR100 and SVHN [39]. See Section E.4 for a visualization of the decision boundaries of normal versus mixup training, which shows that mixup indeed achieves a thicker boundary.

## 3.2 Adversarial training

Here, we compare the boundary thickness of adversarially trained neural networks in different training settings. More specifically, we study the effect of five regularization and data augmentation schemes, including large initial learning rate, $\ell_2$ regularization (weight decay), $\ell_1$ regularization, early stopping, and cutout. We choose a variety of hyper-parameters and plot the robust test accuracy versus thickness. We only choose hyper-parameters such that the natural training accuracy is larger than 90%. We also plot the robust generalization gap versus thickness. See Figure 3a and Figure 3b. We again observe a similar correspondence—the robust generalization gap reduces with increasing thickness.

**Experimental details.** In our experiments, we train a ResNet18 on CIFAR10. In each set of experiments, we only change one parameter. In the standard setting, we follow convention and train with learning rate 0.1, weight decay 5e-4, attack range $\epsilon = 8$ pixels, 10 iterations for each attack, and 2 pixels for the step-size. Then, for each set of experiments, we change one parameter based on the standard setting. For $\ell_2$, $\ell_1$ and cutout, we only use one of them at a time to separate their effects. See Section F.1 for the details of these hyper-parameters. Specifically, see Figure F.1 which shows that all the five regularization and augmentation schemes increase boundary thickness. We train each model for enough time (400 epochs) to let both the accuracy curves and the boundary thickness stabilize, and to filter out the effect of early stopping. In Section F.2, we reimplement the whole procedure with the same early stopping at 120 epochs and learning rate decay at epoch 100. We show that the positive correspondence between robustness and boundary thickness remains the same (see Figure F.2). In Section F.3, we provide an ablation study on the hyper-parameters in measuring thickness

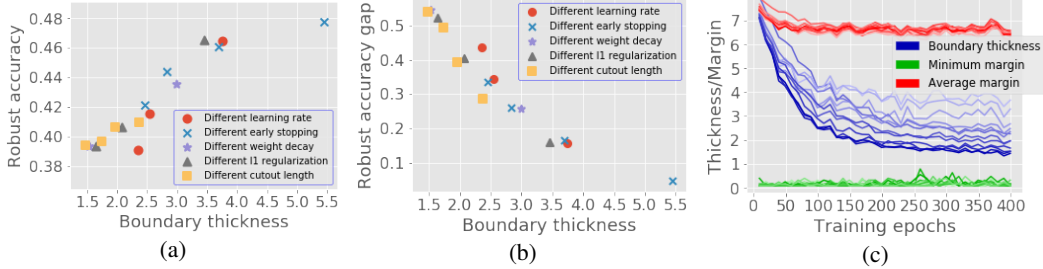

Figure 3: **Adversarial robustness and thickness.** (a) Increasing boundary thickness improves robust accuracy in adversarial training. (b) Increasing boundary thickness reduces overfitting (measured by robust accuracy gap between training and testing). (c) Thickness can differentiate models of different robust levels (dark to light blue), while margin cannot (dark to light red and dark to light green). Results are obtained for ResNet-18 trained on CIFAR10.

| Dataset | Method | Clean | OOD | Black-box | PGD-20 | | |
|---------|--------|-------|-----|-----------|--------|--|--|
| | | | | | 8-pixel | 6-pixel | 4-pixel |
| CIFAR10 | Mixup | **96.0**±0.1 | 78.5±0.4 | 46.3±1.4 | 2.0±0.1 | 3.2±0.1 | 6.3±0.1 |
| | Noisy mixup | 94.4±0.2 | **83.6**±0.3 | **78.0**±1.0 | **11.7**±3.3 | **16.2**±4.2 | **25.7**±5.0 |
| CIFAR100 | Mixup | **78.3**±0.8 | 51.3±0.4 | 37.3±1.1 | 0.0±0.0 | 0.0±0.0 | 0.1±0.0 |
| | Noisy mixup | 72.2±0.3 | **52.5**±0.7 | **60.1**±0.3 | **1.5**±0.2 | **2.6**±0.1 | **6.7**±0.9 |

Table 1: **Mixup and noisy mixup.** The robust test accuracy of noisy mixup is significantly higher than ordinary mixup. Results are reported for ResNet-18 and for the best learning rate in [0.1, 0.03, 0.01].

and again show the same correspondence for the other settings (see Figure F.3). In Section F.4, we provide additional analysis on the comparison between boundary thickness and margin.

### 3.3 Boundary thickness versus margin

Here, we compare margin versus boundary thickness at differentiating robustness levels. See Figure 3c, where we sort the different models shown in Figure 3a by robustness, and we plot their thickness measurements using gradually darker colors. Curves with a darker color represent less robust models. We see that while boundary thickness correlates well with robustness and hence can differentiate different robustness levels, margin is not able to do this.

From (3), we see that computing the margin requires computing the projection $\text{Proj}(x_r, j)$, which is intractable for general nonlinear functions. Thus, we approximate the margin on the direction of an adversarial attack (which is the projection direction for linear classifiers). Another important point here is that we compute the average margin for all samples in addition to the minimum (worst-case) margin in Definition 3. The minimum margin is almost zero in all cases due to the existence of certain samples that are extremely close to the boundary. That is, the standard (widely used) definition of margin performs even worse.

## 4 Applications of Boundary Thickness

While training is not our main focus, our insights motivate new training schemes. At the same time, our insights also aid in explaining the robustness phenomena discovered in some contemporary works, when viewed through the connection between thickness and robustness.

### 4.1 Noisy mixup

Motivated by the success of mixup [16] and our insights into boundary thickness, we introduce and evaluate a training scheme that we call *noisy-mixup*.

**Theoretical justification.** Before presenting noisy mixup, we strengthen the connection between mixup and boundary thickness by stating that the model which minimizes the mixup loss also

achieves optimal boundary thickness in a minimax sense. Specifically, we can prove the following: For a fixed arbitrary integer $c > 1$, the model obtained by mixup training achieves the minimax boundary thickness, i.e., $f_{\text{mixup}}(x) = \arg\max_{f(x)} \min_{(\alpha,\beta)} \Theta(f)$, where the minimum is taken over all possible pairs of $(\alpha, \beta) \in (-1, 1)$ such that $\beta - \alpha = 1/c$, and the max is taken over all prediction functions $f$ such that $\sum_i f(x)_i = 1$. See Section A for the formal theorem statement and proof.

Ordinary mixup thickens decision boundary by mixing different training samples. The idea of noisy mixup, on the other hand, is to thicken the decision boundary between clean samples and arbitrary transformations. This increases the robust performance on OOD images, for example on images that have been transformed using a noise filter or a rotation. Interestingly, pure noise turns out to be good enough to represent such arbitrary transformations. So, while the ordinary mixup training obtains one mixup sample $x$ by linearly combining two data samples $x_1$ and $x_2$, in noisy-mixup, one of the combinations of $x_1$ and $x_2$, with some probability $p$, is replaced by an image that consists of random noise. The label of the noisy image is "NONE." Specifically, in the CIFAR10 dataset, we let the "NONE" class be the $11^{\text{th}}$ class. Note that this method is different than common noise augmentation because we define a new class of pure noise, and we mix it with ordinary samples.

The comparison between the noisy mixup and ordinary mixup training is shown in Table 1. For OOD accuracy, we follow [9] and use both CIFAR-10C and CIFAR-100C. For PGD attack, we use an $\ell_\infty$ attack with 20 steps and with step size being 1/10 of the attack range. We report the results of three different attack ranges, namely 8-pixel, 6-pixel, and 4-pixel. For black-box attack, we use ResNet-110 to generate the transfer attack. The other parameters are the same with the 8-pixel white-box attack. For each method and dataset, we run the training procedures with three learning rates (0.01, 0.03, 0.1), each for three times, and we report the mean and standard deviation of the best performing learning rate. See Section G.1 for more details of the experiment.

From Table 1, we see that noisy mixup significantly improves the robust accuracy of different types of corruptions. Although noisy mixup slightly reduces clean accuracy, the drop of clean accuracy is expected for robust models. For example, we tried adversarial training in the same setting and achieved 57.6% clean accuracy on CIFAR100, which is about 20% drop. In Figure 4, we show that noisy mixup indeed achieves a thicker boundary than ordinary mixup. We use pure noise to represent OOD, but this simple choice already shows a significant improvement in both OOD and adversarial robustness. This opens the door to devising new mechanisms with the goal of increasing boundary thickness to increase robustness against other forms of image imperfections and/or attacks. In Section G.2, we provide further analysis on noisy mixup.

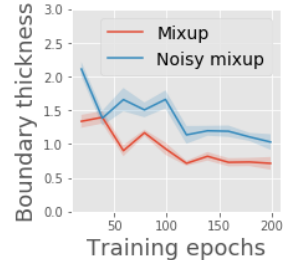

Figure 4: Noisy miuxp thickens the decision boundary.

## 4.2   Explaining robustness phenomena using boundary thickness

**Robustness to image saturation.** We study the connection between boundary thickness and the saturation-based perturbation [40]. In [40], the authors show that adversarial training can bias the neural network towards "shape-oriented" features and reduce the reliance on "texture-based" features. One result in [40] shows that adversarial training outperforms normal training when the saturation on the images is high. In Figure 5a, we show that boundary thickness measured on saturated images in adversarial training is indeed higher than that in normal training.[3]

**A thick boundary reduces non-robust features.** We illustrate the connection to *non-robust features*, proposed by [41] to explain the existence of adversarial examples. The authors show, perhaps surprisingly, that a neural network trained on data that is completely mislabeled through adversarial attacks can achieve nontrivial generalization accuracy on the clean test data (see Section H for the experimental protocols of [41] and their specific way of defining generalization accuracy which we use.) They attribute this behavior to the existence of non-robust features which are essential for generalization but at the same time are responsible for adversarial vulnerability.

We show that the generalization accuracy defined in this sense decreases if the classifier used to generate adversarial examples has a thicker decision boundary. In other words, a thicker boundary removes more non-robust features. We consider four settings in CIFAR10: (1) training without weight

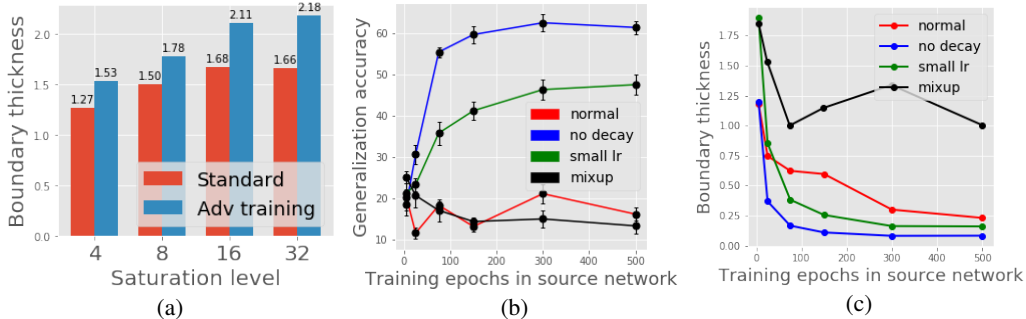

<div align="center">(a)           (b)           (c)</div>

Figure 5: **Explaining robustness phenomena.** (a) The robustness improvement of adversarial training against saturation-based perturbation (studied in [40]) can be explained by a thicker boundary. (b)-(c) Re-implementing the non-robust feature experiment protocol with different training schemes. The two figures show that a thick boundary reduces non-robust features.

decay; (2) training with the standard weight decay 5e-4; (3) training with the standard weight decay but with a small learning rate 0.003 (compared to original learning rate 0.1); and (4) training with mixup. See Figures 5b and 5c for a summary of the results. Looking at these two figures together, we see that an increase in the boundary thickness through different training schemes reduces the generalization accuracy, as defined above, and hence the amount of non-robust features retained. Note that the natural test accuracy of the four source networks cannot explain the difference in Figure 5b, which are 0.943 ("normal"), 0.918 ("no decay"), 0.899 ("small lr"), and 0.938 ("mixup"). For instance, training with no weight decay has the highest generalization accuracy defined in the sense above, but its natural accuracy is only 0.918.

## 5 Conclusions

We introduce boundary thickness, a more robust notion of the size of the decision boundary of a machine learning model, and we provide a range of theoretical and empirical results illustrating its utility. This includes that a thicker decision boundary reduces overfitting in adversarial training, and that it can improve both adversarial robustness and OOD robustness. Thickening the boundary can also reduce boundary tilting and the reliance on "non-robust features." We apply the idea of thick boundary optimization to propose noisy mixup, and we empirically show that using noisy mixup improves robustness. We also show that boundary thickness reduces to margin in a special case, but in general it can be more useful than margin. Finally, we show that the concept of boundary thickness is theoretically justified, by proving that boundary thickness reduces the worst-case boundary tilting and that mixup training achieves the minimax thickness. Having proved a strong connection between boundary thickness and robustness, we expect that further studies can be conducted with thickness and decision boundaries as their focus. We also expect that new learning algorithms can be introduced to increase explicitly boundary thickness during training, in addition to the low-complexity but relatively implicit way of noisy mixup.

## 6 Broader Impact

The proposed concept of boundary thickness can improve our fundamental understanding of robustness in machines learning and neural networks, and thus it provides new ways to interpret black-box models and improve existing robustness techniques. It can help researchers devise novel training procedures to combat data and model corruption, and it can also provide diagnosis to trained machine learning models and newly proposed regularization or data augmentation techniques.

The proposed work will mostly benefit safety-critical applications, e.g., autonomous driving and cybersecurity, and it may also make AI-based systems more reliable to natural corruptions and imperfections. These benefits are critical because there is usually a gap between the performance of learning-based systems on well-studied datasets and in real-life scenarios.

We will conduct further experimental and theoretical research to understand the limitations of boundary thickness, both as a metric and as a general guideline to design training procedures. We also believe that the machine learning community needs to conduct further research to enhance the fundamental understandings of the structure of decision boundaries and the connection to robustness. For instance, it is useful to design techniques to analyze and visualize decision boundaries during both training (e.g., how the decision boundary evolves) and testing (e.g., how to find defects in the boundary.) We believe this will benefit both the safety and accountability of learning-based systems.

**Acknowledgments**

We would like to thank Zhewei Yao, Tianjun Zhang and Dan Hendrycks for their valuable feedback. Michael W. Mahoney would like to acknowledge the UC Berkeley CLTC, ARO, IARPA, NSF, and ONR for providing partial support of this work. Kannan Ramchandran would like to acknowledge support from NSF CIF-1703678 and CIF-2002821. Joseph E. Gonzalez would like to acknowledge supports from NSF CISE Expeditions Award CCF-1730628 and gifts from Amazon Web Services, Ant Group, CapitalOne, Ericsson, Facebook, Futurewei, Google, Intel, Microsoft, Nvidia, Scotiabank, Splunk and VMware. Our conclusions do not necessarily reflect the position or the policy of our sponsors, and no official endorsement should be inferred.

## Footnotes

[1]https://github.com/nsfzyzz/boundary_thickness

[2]The models in Figure 2 are from https://github.com/kuangliu/pytorch-cifar/blob/master/models/resnet.py.

[3]We use the online implementation in https://github.com/PKUAI26/AT-CNN.

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
