[Supplementary Material]

# Supplementary Materials of "Boundary Thickness and Robustness in Learning Models"

## A    Mixup Increases Thickness

In this section, we show that mixup as well as the noisy mixup scheme studied in Section 4.1 both increase boundary thickness.

Recall that $x_r$ and $x_s$ in (1) are not necessaraily from the training data. For example, $x_r$ and/or $x_s$ can be the noisy samples used in the noisy mixup (Section 4.1). We make the analysis more general here because in different extensions of mixup [9, 16, 30], the mixed samples can either come from the training set, from adversarial examples constructed from the training set, or from carefully augmented samples using various forms of image transforms.

We consider binary classification and study the unnormalized empirical version of (1) defined as follows:

$$\Theta(f, \alpha, \beta) := \sum_{(x_i, x_j) \text{ s.t. } y_i \neq y_j} \|x_r - x_s\| \int_{t \in [0,1]} \mathbf{I}\{\alpha < g_{01}(x(t)) < \beta\} dt, \tag{7}$$

where the expectation in (1) is replaced by its empirical counterpart. We now show that the function which achieves the minimum mixup loss is also the one that achieves minimax thickness for binary classification.

**Proposition A.1** (Mixup Increases Boundary Thickness). *For binary classification, suppose there exists a function $f_{mixup}(x)$ that achieves exactly zero mixup loss, i.e., on all possible pairs of points $((x_r, y_r), (x_s, y_s))$, $f_{mixup}(\lambda x_r + (1 - \lambda)x_s) = \lambda y_r + (1 - \lambda)y_s$ for all $\lambda \in [0,1]$. Then, for an arbitrary fixed integer $c > 1$, $f_{mixup}(x)$ is also a solution to the following minimax problem:*

$$\arg \max_f \min_{(\alpha, \beta)} \Theta(f, \alpha, \beta), \tag{8}$$

*where the boundary thickness $\Theta$ is defined in Eqn. (7), the maximization is taken over all the 2D functions $f(x) = [f(x)_0, f(x)_1]$ such that $f(x)_0 + f(x)_1 = 1$ for all $x$, and the minimization is taken over all pairs of $\alpha, \beta \in (-1, 1)$ such that $\beta - \alpha = 1/c$.*

*Proof.* See Section B.4 for the proof.  □

**Remark 2** (Why Mixup is Preferred among Different Thick-boundary Solutions). Here, we only prove that mixup provides one solution, instead of the only solution. For example, between two samples $x_r$ and $x_s$ that have different labels, a piece-wise 2D linear mapping that oscillates between $[0, 1]$ and $[1, 0]$ for more than once can achieve the same thickness as that of a linear mapping. However, a function that exhibits unnecessary oscillations becomes less robust and more sensitive to small input perturbations. Thus, the linear mapping achieved by mixup is preferred. According to [16], mixup can also help reduce unnecessary oscillations.

**Remark 3** (Zero Loss in Proposition A.1). Note that the function $f_{mixup}(x)$ in the proposition is the one that perfectly fits the mixup augmented dataset. In other words, the theorem above needs $f_{mixup}(x)$ to have "infinite capacity," in some sense, to match perfectly the response on line segments that connect pairs of points $(x_r, x_s)$. If such $f_{mixup}(x)$ does not exist, it is unclear if an approximate solution achieves minimax thickness, and it is also unclear if minimizing the cross-entropy based mixup loss is exactly equivalent to minimizing the minimax boundary thickness for the same loss value. Nonetheless, our experiments show that mixup consistently achieves thicker decision boundaries than ordinary training (see Figure 2a).

## B    Proofs

### B.1    Proof of Proposition 2.1

Choose $(x_r, x_s)$ so that $x_s - x_r = cw$. The thickness of $f$ defined in (1) becomes

$$\Theta(f, \alpha, \beta, x_s, x_r) = c\|w\| \mathbb{E}_p \left[ \int_0^1 \mathbf{I}\{\alpha < g_{01}(x(t)) < \beta\} dt \right]. \tag{9}$$

Define a substitute variable $u$ as:

$$u = tw^\top x_r + (1-t)w^\top x_s + b. \tag{10}$$

Then,

$$du = (w^\top x_r - w^\top x_s)dt = w^\top(-cw)dt = -c\|w\|^2 dt. \tag{11}$$

Further,

$$f(tx_r + (1-t)x_s)_0 - f(tx_r + (1-t)x_s)_1 = 2\sigma(tw^\top x_r + (1-t)w^\top x_s + b) - 1 = 2\sigma(u) - 1 = \tilde{g}(u). \tag{12}$$

Thus,

$$\Theta(f, \alpha, \beta, p) \overset{(a)}{=} c\|w\|\mathbb{E}\left[\int_0^1 \mathbf{I}\{\alpha < f(tx_r + (1-t)x_s)_0 - f(tx_r + (1-t)x_s)_1 < \beta\}dt\right]$$

$$\overset{(b)}{=} \mathbb{E}\left[\int_{w^\top x_s + b}^{w^\top x_r + b} \mathbf{I}\{\alpha < \tilde{g}(u) < \beta\}\left(-\frac{1}{\|w\|}\right)du\right] \tag{13}$$

$$\overset{(c)}{=} \frac{1}{\|w\|}\mathbb{E}\left[\int_{w^\top x_r + b}^{w^\top x_s + b} \mathbf{I}\{\alpha < \tilde{g}(u) < \beta\}du\right],$$

where $(a)$ holds because $g_{01}(x(t)) = f(x(t))_0 - f(x(t))_1 = f(tx_r + (1-t)x_s)_0 - f(tx_r + (1-t)x_s)_1$, $(b)$ is from substituting $u = tw^\top x_r + (1-t)w^\top x_s + b$ and $du = -c\|w\|^2 dt$, and $(c)$ is from switching the upper and lower limit of the integral to get rid of the negative sign. Recall that $\tilde{g}(u)$ is a monotonically increasing function in $u$. Thus,

$$\Theta(f, \alpha, \beta, p) = \frac{1}{\|w\|}\int_{w^\top x_r + b}^{w^\top x_s + b} \mathbf{I}\{\tilde{g}^{-1}(\alpha) < u < \tilde{g}^{-1}(\beta)\}du$$

$$= \frac{1}{\|w\|}\mathbb{E}\left[\min(\tilde{g}^{-1}(\beta), w^\top x_s + b) - \max(\tilde{g}^{-1}(\alpha), w^\top x_r + b)\right]. \tag{14}$$

Further, if $[\alpha, \beta]$ is contained in $[g_{01}(x_r), g_{01}(x_s)]$, we have $\tilde{g}^{-1}(\beta) < \tilde{g}^{-1}(g_{01}(x_s)) = \tilde{g}^{-1}(\tilde{g}(w^\top x_s + b)) = w^\top x_s + b$, and similarly, $\tilde{g}^{-1}(\alpha) > w^\top x_r + b$, and thus

$$\Theta(f, \alpha, \beta) = (\tilde{g}^{-1}(\beta) - \tilde{g}^{-1}(\alpha))/\|w\|. \tag{15}$$

## B.2 Proof of Proposition 2.2

The conclusion holds if $\|x_r - x_s\|\int_0^1 \mathbf{I}\{\alpha < g_{ij}(x(t)) < \beta\}dt$ equals the $\ell_2$ distance from $x_r$ to its projection $\text{Proj}(x, j)$ for $\alpha = 0$ and $\beta = 1$. Note that when $x = x_s$, $g_{ij}(x) = 0$, because $x_s = \text{Proj}(x_r, j)$. From the definition of projection, i.e., $\text{Proj}(x, j) = \arg\min_{x' \in S(i_x, j)} \|x' - x\|$, we have that for all points $x$ on the segment from $x_r$ to $x_s$, $x_s$ is only point with $g_{ij}(x) = 0$. Otherwise, $x_s$ is not the projection. Therefore, all points $x$ on the segment satisfy $g_{ij}(x) > 0 = \alpha$. Since $f$ is the output after the softmax layer, $g_{ij}(x) = f(x)_i - f(x)_j < 1 = \beta$. Thus, the indicator function on the left-hand-side of (4) takes value 1 always, and the integration reduces to calculating the distance from $x_r$ to $x_s$.

## B.3 Proof of Proposition 2.3

We rewrite the definition of $T(u)$ as

$$T(u) := \min_{v \text{ s.t. } \|v\| = u \text{ and } y_i v^\top x_i \geq 1, \forall i} \text{Cosine Similarity}(v, w^*), \tag{16}$$

where

$$\text{Cosine Similarity}(v, w^*) := |v^\top w^*|/(\|v\| \cdot \|w^*\|). \tag{17}$$

To prove $T(u)$ is a non-increasing function in $u$, we consider arbitrary $u_1, u_2$ so that $u_1 > u_2 \geq \|w^*\|$, and we prove $T(u_1) \leq T(u_2)$.

First, consider $T(u_2)$. Denote by $w_2$ the linear classifier that achieves the minimum value in the RHS of (16) when $u = u_2$. From definition, $u_2 = \|w_2\|$. Now, if we increase the norm of $w_2$ to obtain a new classifier $w_1 = \frac{u_1}{u_2}w_2$, it still satisfies the constraint $y_i w_1 x_i \geq 1, \forall i$ because

$$y_i w_1 x_i = \frac{u_1}{u_2}y_i w_2 x_i \geq \frac{u_1}{u_2} > 1. \tag{18}$$

Thus, $w_1 = \frac{u_1}{u_2} w_2$ satisfies the constraints in (16) for $u = \|w_1\| = u_1$, and being a linear scaling of $w_2$, it has the same cosine similarity score with $w^\star$ (17), which means the worst-case tilting $T(u_1)$ should be smaller or equal to the tilting of $w_1$.

## B.4 Proof of Proposition A.1

We can rewrite (7) using

$$\Theta(f, \alpha, \beta) := \sum_{(x_i, x_j) \text{ s.t. } y_i \neq y_j} \Theta_{1D}(f, \alpha, \beta, x_r, x_s), \tag{19}$$

where $\Theta_{1D}(f, \alpha, \beta, x_r, x_s)$ denotes the thickness measured on a single segment, i.e.,

$$\Theta_{1D}(f, \alpha, \beta, x_r, x_s) := \|x_r - x_s\| \int_{t \in [0,1]} \mathbf{I}\{\alpha < g_{01}(x(t)) < \beta\} dt, \tag{20}$$

where recall that $g_{01}(x) = f(x)_0 - f(x)_1$ and $x(t) = t x_r + (1 - t) x_s$.

Since the proposition is stated for the sum on all pairs of data, we can focus on the proof of an arbitrary pair of data $(x_r, x_s)$ such that $y_r \neq y_s$.

Consider any 2D decision function $f(x) = [f(x)_0, f(x)_1]$ such that $f(x)_0 + f(x)_1 = 1$ (i.e., $f(x)$ is a probability mass function). In the following, we consider the restriction of $f(x)$ on a segment $(x_r, x_s)$, which we denote as $f_{(x_r, x_s)}(x)$. Then, the proof relies on the following lemma, which states that the linear interpolation scheme in mixup training does maximize the boundary thickness on the segment.

**Lemma B.1.** *For any arbitrary fixed integer $c > 0$, the linear function*

$$f_{lin}(t x_r + (1 - t) x_s) = [t, 1 - t], t \in [0, 1], \tag{21}$$

*defined for a given segment $(x_r,\ x_s)$ optimizes $\Theta_{ID}(f, \alpha, \beta, x_r, x_s)$ in (20) in the following minimax sense,*

$$f_{lin}(x) = \arg \max_{f_{(x_r, x_s)}(x)} \min_{(\alpha, \beta)} \Theta_{ID}(f_{(x_r, x_s)}(x), \alpha, \beta, x_r, x_s), \tag{22}$$

*where the maximization is over all the 2D functions $f_{(x_r, x_s)}(x) = [f_{(x_r, x_s)}(x)_0, f_{(x_r, x_s)}(x)_1]$ such that the domain is restricted to the segment $(x_r,\ x_s)$ and such that $f_{(x_r, x_s)}(x)_0, f_{(x_r, x_s)}(x)_1 \in [0, 1]$ and $f_{(x_r, x_s)}(x)_0 + f_{(x_r, x_s)}(x)_1 = 1$ for all $x$ on the segment, and the minimization is taken over all pairs of $\alpha, \beta \in (-1, 1)$ such that $\beta - \alpha = 1/c$.*

*Proof.* See Section B.5 for the proof. $\square$

Now, Proposition A.1 follows directly from Lemma B.1.

## B.5 Proof of Lemma B.1

In this proof, we simplify the notation and use $f(x)$ to denote $f_{(x_r, x_s)}(x)$ which represents $f(x)$ restricted to the segment $(x_r, x_s)$. This simple notation does not cause any confusion because we restrict to the segment $(x_r, x_s)$ in this proof.

We can simplify the proof by viewing the optimization over functions $f(x)$ on the fixed segment $(x_r, x_s)$ as optimizing over the functions $h(t) = f(x(t))_1 - f(x(t))_0$ on $t \in [0, 1]$, where $x(t) = t x_r + (1 - t) x_s$.

Thus, we only need to find the function $f(x)$, when viewed as a one-dimensional function $h(t) = f(x(t))_1 - f(x(t))_0 = f(t x_r + (1-t) x_s)_1 - f(t x_r + (1-t) x_s)_0$, that solves the minimax problem (22) for the thickness defined as:

$$\begin{aligned}
\Theta_{1D}(f(t)) &= \|x_r - x_s\| \int_{t \in [0,1]} \mathbf{I}\{\alpha < h(t) < \beta\} dt \\
&= \|x_r - x_s\| |h^{-1}((\alpha, \beta))|,
\end{aligned} \tag{23}$$

where $h^{-1}$ is the inverse function of $h(t)$. Note that $h^{-1}((\alpha, \beta)) \subset [0, 1]$. To prove the result, we only need to prove that the linear function $h_{\text{lin}}(t) = 2t - 1$, which is obtained from $h(t) = f(tx_r + (1 - t)x_s)_1 - f(tx_r + (1 - t)x_s)_0$ for $f_{\text{lin}}(tx_r + (1 - t)x_s) = [t, 1 - t]$ defined in (21), solves the minimax problem

$$\arg\max_{h(t)} \min_{(\alpha, \beta)} \Theta_{\text{1D}}(h(t)) = \arg\max_{h(t)} \min_{(\alpha, \beta)} |h^{-1}((\alpha, \beta))|, \tag{24}$$

where the maximization is taken over all $h(t)$, and the minimization is taken over all pairs of $\alpha, \beta \in (-1, 1)$ such that $\beta - \alpha = 1/c$, for a fixed integer $c > 1$.

Now we prove a stronger statement.

**Stronger statement:**

$$\min_{(\alpha, \beta)} |h^{-1}((\alpha, \beta))| \leq \frac{\beta - \alpha}{2}, \tag{25}$$

when the minimization is taken over all $\alpha, \beta$ such that $\alpha - \beta = 1/c$, and for any measurable function $h(t) : [0, 1] \to [-1, 1]$.

If we can prove this statement, then, since $h_{\text{lin}}(t) = 2t - 1$ always achieves $h^{-1}((\alpha, \beta)) = \frac{\beta + 1}{2} - \frac{\alpha + 1}{2} = \frac{\beta - \alpha}{2}$, it is indeed the minimax solution of (24).

We prove the stronger statement above by contradiction. Suppose that the statement is not true, i.e., for any $\alpha$ and $\beta$ such that $\beta - \alpha = 1/c$, we always have

$$|h^{-1}((\alpha, \beta))| > \frac{\beta - \alpha}{2} = \frac{1}{2c}. \tag{26}$$

Then, the pre-image of $[-1, 1]$ satisfies

$$\begin{aligned}
1 &\geq |h^{-1}([-1, 1])| \\
&= \sum_{i=-c+1}^{c} \left| h^{-1}\left( \left(\frac{i-1}{c}, \frac{i}{c}\right) \right) \right| \\
&\overset{(a)}{>} 2c \cdot \frac{1}{2c} = 1,
\end{aligned} \tag{27}$$

where the last inequality holds because of the inequality (26). This is clearly a contradiction, which means that the stronger statement is true.

## C  A Chessboard Toy Example

In this section, we use a chessboard example in low dimensions to show that nonrobustness arises from thin boundaries. Being a toy setting, this is limited in the generality, but it can visually demonstrate our main message.

In Figure C.1, the two subfigures shown on the first row represent the chessboard dataset and a robust 2D function that correctly classifies the data. Then, we project the 2D chessboard data to a 100-dimensional space by padding noise to each 2D point. In this case, the neural network can still learn the chessboard pattern and preserve the robust decision boundary (see the 3D top-down view on the left part of the second row which contains the chessboard pattern).

However, if we randomly perturb each square of samples in the chessboard in the 3rd dimension (the $z$ axis) to change the space between these squares, such that the boundary has enough space on the $z$-axis to partition the two opposite classes, the boundary changes to a non-robust one instead (see the right part on the second row of Figure C.1). The shift value on the $z$ axis is $0.05$ which is much smaller than the distance between two adjacent squares, which is $0.6$. The data are not linearly separable on the $z$-axis because each square on the chessboard is randomly shifted up or down independently of other squares.

A more interesting result can be obtained by varying the shift on the $z$ axis from 0.01 to 0.08. See the third row of Figure C.1. The network undergoes a sharp transition from using robust decision boundaries to using non-robust ones. This is consistent with the main message shown in Figure 1,

| 2D chessboard data | 2D classifier |
|---|---|

**3D Robust classifier**

| Front view | Top-down view |
|---|---|

**3D Non-robust classifier**

| Front view | Top-down view |
|---|---|

**Interpolation between robust and non-robust classifier**

Figure C.1: **Chessboard toy example.** The 3D visualizations above use the color map that ranges from yellow (value 1) to purple (value 0) to illustrate the predicted probability $\Pr(y = 1|x) \in [0, 1]$ in binary classification. Each 3D figure draws 17 level sets of different colors from 0 to 1.

**First row:** The 2D chessboard dataset with two classes and a 2D classifier that learns the correct pattern.

**Second row:** 3D visualization of decision boundaries of two different classifiers. **(left)** A classifier that uses robust $x$ and $y$ directions to classify, which preserves the complex chessboard pattern (see the top-down view which contains a chessboard pattern.) **(right)** A classifier that uses the non-robust direction $z$ to classify. When the separable space on the non-robust direction is large enough, the thin boundary squeezes in and generates a simple but non-robust function.

**Third row:** Visulization of the decision boundary as we interpolate between the robust and non-robust classifiers. There is a sharp transition from the fourth to the fifth figure.

i.e., that neural networks tend to generate thin and non-robust boundaries to fit in the narrow space between opposite classes on the non-robust direction, while a thick boundary mitigates this effect. On the third row, from left to right, the expanse of the data on $z$-axis increases, allowing the network to use only the $z$-axis to classify.

**Details of 3D figure generation:** For the visualization in Figure C.1, each 3D figure is generated by plotting 17 consecutive level sets of neural network prediction values (after the softmax layer) from 0 to 1. The prediction on each level set is the same, and each level set is represented by a colored continuous surface in the 3D figure. The yellow end of the color bar represents a function value of 1, and the purple end represents 0. The visualization is displayed in a 3D orthogonal subspace of the whole input space. The three axes are the first three vectors in the natural basis. They represent the $x$ and $y$ directions that contain the chessboard pattern, and the $z$ axis that contains the direction of shift values.

**Details of the chessboard toy example:** The chessboard data contains 2 classes of 2D points arranged in $9 \times 9 = 81$ squares. Each square contains 100 randomly generated 2D points uniformly distributed in the square. The length of each square is 0.4, and the separation between two adjacent squares is 0.6. The shift direction (up or down) and value on the $z$-axis are the same for all 2D points in a single square, and the shift value is much smaller than 1 (which is the distance between the centers of two squares). See the third row on Figure C.1 for different shift values ranging from 0.01 to 0.08. The shift value is, however, independent across different squares, i.e., these squares cannot be easily separated by a linear classifier using information on the $z$-axis only. The classifier is a neural network with 9 fully-connected layers and a residual link on each layer. The training has 100 epochs, an initial learning rate of 0.003, batch size 128, weight decay 5e-4, and momentum 0.9.

Figure D.1: **Thickness and boundary tilting.** (a) The cosine similarity between the gradient direction and $x_s - x_r$ generalizes the measurement of "boundary tilting" to nonlinear functions. (b) Boundary tilting can be mitigated by using a thick decision boundary.

## D    A Thick Boundary Mitigates Boundary Tilting

In this section, we generalize the observation of Proposition 2.3 to nonlinear classifiers. Recall that in Proposition 2.3, we use Cosine Similarity $(w, w^*) = |w^\top w^*|/(\|w\| \cdot \|w^*\|)$ between the classifier $w$ and the max-margin classifier $w^*$ to measure boundary tilting. To measure boundary tilting in the nonlinear case, we use $x_1 - x_2$ of random sample pairs $(x_1, x_2)$ from the training set to replace the normal direction of the max-margin solution $w^*$, and use $\nabla g_{ij}(x) = \nabla (f(x)_i - f(x)_j)$ to replace the normal direction of a linear classifier $w$, where $i, j$ are the predicted labels of $x_1$ and $x_2$, respectively, and $x$ is a random point on the line segment $(x_1, x_2)$. Then, the cosine similarity generalizes to

$$\text{Average Cosine Similarity} = \mathbb{E}_{(x_1, x_2) \sim \text{training distribution s.t. } i \neq j} \left[ \frac{|\langle x_1 - x_2, \nabla_x g_{ij}(x)\rangle|}{\|x_1 - x_2\| \|\nabla_x g_{ij}(x)\|} \right]. \quad (28)$$

In Figure D.1a, we show the intuition underlying the use of (28). The smaller the cosine similarity is, the more severe the impact of boundary tilting becomes.

We also measure boundary tilting in various settings of adversarial training, and we choose the same set of hyper-parameters that are used to generate Figure 3. See the results in Figure D.1b. When measuring cosine similarity, we average the results over 6400 training sample pairs. From the results shown in Figure D.1b, a thick boundary mitigates boundary tilting by increasing the cosine similarity.

## E    Additional Experiments on Non-adversarial Training

In this section, we provide more details and additional experiments extending the results of Section 3.1 on non-adversarially trained neural networks. We demonstrate that a thick boundary improves OOD robustness when the thickness is measured using different choices of hyper-parameters. We also show that the same conclusion holds on two other datasets, namely CIFAR100 and SVHN, in addition to CIFAR10 used in the main paper.

### E.1    Details of measuring boundary thickness

Boundary thickness is calculated by integrating on the segments that connect a sample with its corresponding adversarial sample. We find the adversarial sample by using an $\ell_2$ PGD attack of size 1.0, step size 0.2, and number of attack steps 20. We measure both thickness and margin on the normalized images in CIFAR10, which introduces a multiplicity factor of approximately 5 when using the standard deviations $(0.2023, 0.1994, 0.2010)$, respectively, for the RGB channels compared to measuring thickness on unnormalized images.

To compute the integral in (1), we connect the segment from $x_r$ to $x_s$ and evaluate the neural network response on 128 evenly spaced points on the segment. Then, we compute the cumulative $\ell_2$ distance

Figure E.1: **Thickness on random sample pairs.** Measuring the boundary thickness in the same experimental setting as Figure 2a, but on pairs of random samples. The trend that mixup > normal training > training without weight decay remains the same.

of the parts on this segment for which the prediction value is between $(\alpha, \beta)$, which measures the distance between two level sets $g_{ij}(x) = \alpha$ and $g_{ij}(x) = \beta$ on this segment (see equation (1)). Finally, we report the average thickness obtained by repeated runs on 320 segments, i.e., 320 random samples with their adversarial examples.

### E.2 Comparing different measuring methods: tradeoff between complexity and accuracy

In this section, we discuss the choice of distribution $p$ when selecting segments $(x_r, x_s)$ to measure thickness. Recall that in the main paper, we choose $x_s$ as an adversarial example of $x_r$. Another way, which is computationally cheaper, is to measure thickness on the segment directly between pairs of samples in the training dataset, i.e., sample $x_r$ randomly from the training data, and sample $x_s$ as a random data point with a different label.

Although computationally cheaper, this way of measuring boundary thickness is more prone to the "boundary-tilting" effect, because the connection between a pair of samples is not guaranteed to be orthogonal to the decision boundary. Thus, the boundary tilting effect can inflate the value of boundary thickness. This effect only happens when we measure thickness on pairs of samples instead of measuring it in the adversarial direction, which we have shown to be able to mitigate boundary tilting when the thickness is large (see Section D).

In Figure E.1, we show how this method affects the measurement of thickness. The thickness is measured for the same set of models and training procedures as those shown in Figure 2a, but on random segments that connect pairs of samples. We use $\alpha = 0$ and $\beta = 0.75$ to match Figure 2a. In Figure E.1, although the trend remains the same (i.e., mixup>normal>training without weight decay), all the measurement values of boundary thickness become much bigger than that of Figure 2a, indicating boundary tilting in all the measured networks.

**Remark 4** (An Oscillating 1D Example Motivates the Adversarial Direction). Obviously, the distribution $p$ in Definition 1 is vital in dictating robustness. Similar to Remark 2, one can consider an example of 2D piece-wise linear mapping $f(x) = [f(x)_0, f(x)_1]$ on a segment $(x_r, x_s)$ that oscillates between the response $[0, 1]$ and $[1, 0]$. If one measures the thickness on this particular segment, the measured thickness remains the same if the number of oscillations increases in the piece-wise linear mapping, but the robustness reduces with more oscillations. Thus, the example motivates the measurement on the direction of an adversarial attack, because an adversarial attack tends to find the closest "peak" or "valley" and can thus faithfully recover the correct value of boundary thickness unaffected by the oscillation.

### E.3 Ablation study

In this section, we provide an extensive ablation study on the different choices of hyper-parameters used in the experiments. We show that our main conclusion about the positive correlation between robustness and thickness remains the same for a wide range of hyper-parameters obviating the need to fine-tune these. We study the adversarial attack direction used to measure thickness, the parameters $\alpha$ and $\beta$, as well as reproducing the results on two other datasets, namely CIFAR100 and SVHN, in addition to CIFAR10.

(a) Results on CIFAR10 with a large attack $\epsilon = 2.0$

(b) Results on CIFAR10 with a small attack $\epsilon = 0.6$

Figure E.2: **Ablation study on different attack sizes.** Re-implementing the measurements in Figure 2a using a larger or a smaller adversarial attack.

### E.3.1 Different choices of adversarial attack in measuring boundary thickness

To measure boundary thickness on the adversarial direction, we have to specify a way to implement the adversarial attack. To generate Figure 2a, we used $\ell_2$ attack with attack range $\epsilon = 1.0$, step size 0.2, and number of attack steps 20. We show that the results and more importantly our conclusions do not change by perturbing $\epsilon$ a little. See Figure E.2 and compare it with the corresponding results presented in Figure 2a. We see that the change in the size of the adversarial attack does not alter the trend. However, the measured thickness value does shrink if the $\epsilon$ becomes too small, which is expected.

### E.3.2 Different choices of $\alpha$ and $\beta$ in measuring boundary thickness

In this subsection, we present an ablation study on the choice of hyper-parameters $\alpha$'s and $\beta$ in (1). We show that the conclusions in Section 3.1 remain unchanged for a wide range of choices of $\alpha$ and $\beta$. See Figure E.3. From the results, we can see that the trend remains the same, i.e., mixup>normal training>training without weight decay. However, when $\alpha$ and $\beta$ become close to each other, the magnitude of boundary thickness also reduces, which is expected.

**Remark 5** (Choosing the Best Hyper-parameters). From Proposition 2.2, we know that the margin has particular values of the hyper-parameters $\alpha = 0$ and $\beta = 1$. Allowing different values of these hyper-parameters allows us the flexibility to better capture the robustness than margin. The best choices of these hyper-parameters might be different for different neural networks, and ideally one could do small validation based studies to tune these hyper-parameters, but our ablation study in this section shows that for a large regime of values, the exact search for the best choices is not required. We noticed, for example, setting $\alpha = 0$ and $\beta = 0.75$ works well in practice, and much better than the standard definition of margin that has been equated to robustness in past studies.

**Remark 6** (Choosing Asymmetric $\alpha$ and $\beta$). We use asymmetric parameters $\alpha = 0$ and $\beta > 0$ mainly because, due to symmetry, the measured thickness when $(\alpha, \beta) = (0, x)$ is half in expectation of that when $(\alpha, \beta) = (-x, x)$.

We have discussed alternative ways of adversarial attacks to measure boundary thickness on sample pairs in Section E.2. For completeness, we also do an ablation study for choice of hyper-parameters $\alpha$ and $\beta$ for this case. The results in Figure E.4, and this study also reinforces the same conclusion – that the particular choice of $\alpha$, $\beta$ matters less than the fact that they are not set to 0 and 1 respectively.

### E.3.3 Additional datasets

We repeat the experiments in Section 3.1 on two more datasets, namely CIFAR100 and SVHN. See Figure E.5. In this figure, we used the same experimental setting as in Section 3.1, except that we

(a) Results on CIFAR10 for $\alpha = 0$ and $\beta = 0.9$

(b) Results on CIFAR10 for $\alpha = 0$ and $\beta = 0.8$

(c) Results on CIFAR10 for $\alpha = 0$ and $\beta = 0.7$

(d) Results on CIFAR10 for $\alpha = 0$ and $\beta = 0.6$

Figure E.3: **Ablation study on different $\alpha$ and $\beta$.** Re-implementing the measurements in Figure 2a for different choices of $\alpha$ and $\beta$ in Eqn.(1).

train with a different initial learning rate 0.01 on SVHN, following convention. We reach the same conclusion as in Section 3.1, i.e., that mixup increases boundary thickness, while training without weight decay reduces boundary thickness.

### E.4 Visualizing neural network boundaries

In this section, we show a qualitative comparison between a neural network trained using mixup and another one trained in a standard way without mixup. See Figure E.6. In the left figure, we can see that different level sets are spaced apart, while the level sets in the right figure are hardly distinguishable. Thus, the mixup model has a larger boundary thickness than the naturally trained model for this setting.

For the visualization shown in Figure E.6, we use 17 different colors to represent the 17 level sets. The origin represents a randomly picked CIFAR10 image. The $x$-axis represents a direction of adversarial perturbation found using the projected gradient descent method [6]. The $y$-axis and the $z$-axis represent two random directions that are orthogonal to the $x$ perturbation direction. Each CIFAR10 input image has been normalized using standard routines during training, e.g., using the standard deviations $(0.2023, 0.1994, 0.2010)$, respectively, for the RGB channels, so the scale of the figure may not represent the true scale in the original space of CIFAR10 input images.

(a) Results on CIFAR10 for $\alpha = 0$ and $\beta = 0.9$

(b) Results on CIFAR10 for $\alpha = 0$ and $\beta = 0.8$

(c) Results on CIFAR10 for $\alpha = 0$ and $\beta = 0.7$

(d) Results on CIFAR10 for $\alpha = 0$ and $\beta = 0.6$

Figure E.4: **Ablation study on different $\alpha$ and $\beta$ for thickness measured on random sample pairs.** Re-implementing the measurements in Figure E.1 for different choices of $\alpha$ and $\beta$ in Eqn.(1).

(a) Results on CIFAR100

(b) Results on SVHN

Figure E.5: **Ablation study on more datasets.** Re-implementing the measurements in Figure 2a on for two other datasets CIFAR100 and SVHN.

Figure E.6: **Visualization of mixup.** A mixup model has a thicker boundary than the same model trained using standard setting, because the level sets represented by different colors are more separate in mixup than the standard case.

# F  Additional Experiments on Adversarial Training

In this section, we provide additional details and analyses for the experiments in Section 3.2 on adversarially trained neural networks. We demonstrate that a thick boundary improves adversarial robustness for wide range of hyper-parameters, including those used during adversarial training and those used to measure boundary thickness.

## F.1  Details of experiments in Section 3.2

We use ResNet-18 on CIFAR-10 for all the experiments in Section 3.2. We first choose a standard setting that trains with learning rate 0.1, no learning rate decay, weight decay 5e-4, attack range $\epsilon = 8$ pixels, 10 iterations for each attack, and 2 pixels for the step-size. Then, for each set of experiments, we change one parameter based on the standard setting. We tune the parameters to achieve a natural training accuracy larger than 90%. For the experiment on early stopping, we use a learning rate 0.01 instead of 0.1 to achieve 90% training accuracy. We train the neural network for 400 epochs without learning rate decay to filter out the effect of early stopping. The results with learning rate decay and early stopping are reported in Section F.2 which show the same trend.

When measuring boundary thickness, we select segments on the adversarial direction, and we find the adversarial direction by using an $\ell_2$ PGD attack of size $\epsilon = 2.0$, step size 0.2, and number of attack steps 20.

| Changed parameter | Learning rate | Weight decay | L1 | Cutout | Early stopping |
|---|---|---|---|---|---|
| Learning rate | 3e-3 | 5e-4 | 0 | 0 | None |
|  | 1e-2 | 5e-4 | 0 | 0 | None |
|  | 3e-2 | 5e-4 | 0 | 0 | None |
| Weight decay | 1e-1 | 0e-4 | 0 | 0 | None |
|  | 1e-1 | 1e-4 | 0 | 0 | None |
| L1 | 1e-1 | 0 | 5e-7 | 0 | None |
|  | 1e-1 | 0 | 2e-6 | 0 | None |
|  | 1e-1 | 0 | 5e-6 | 0 | None |
| Cutout | 1e-1 | 0 | 0 | 4 | None |
|  | 1e-1 | 0 | 0 | 8 | None |
|  | 1e-1 | 0 | 0 | 12 | None |
|  | 1e-1 | 0 | 0 | 16 | None |
| Early stopping | 1e-2 | 5e-4 | 0 | 0 | 50 |
|  | 1e-2 | 5e-4 | 0 | 0 | 100 |
|  | 1e-2 | 5e-4 | 0 | 0 | 200 |
|  | 1e-2 | 5e-4 | 0 | 0 | 400 |

Table F.1: **Hyper-parameters in Section 3.2.** The table reports the hyper-parameters used to obtain the results in Figure 3a and 3b for adversarial training.

Note that boundary thickness indeed increases with heavier regularization or data augmentation. See Figure F.1 on the thickness of the models trained with the parameters reported in Table F.1.

Figure F.1: **Regularization and data augmentation increase thickness.** The five commonly used regularization and data augmentation schemes studied in Section 3.2 all increase boundary thickness.

(a)                                                    (b)

Figure F.2: **Adversarial training with learning rate decay.** Reimplementing the experimental protocols that obtained Figure 3 using learning rate decay. The hyper-parameters are shown in Table F.2

## F.2 Adversarial training with learning rate decay

Here, we reimplement the experiments shown in Figure 3a and 3b but with learning rate decay and early stopping, which is reported by [33] to improve the robust accuracy of adversarial training. We still use ResNet-18 on CIFAR-10. However, instead of training for 400 epochs, we train for only 120 epochs, with a learning rate decay of 0.1 at epoch 100. The adversarial training still uses 8-pixel PGD attack with 10 steps and step size 2 pixel.

The set of training hyper-parameters that we use are shown in Table F.2. Similar to Figure 3, we tune hyper-parameters such that the training accuracy on natural data reaches 90%. The results are reported in Figure F.2. We do not separately test early stopping because all experiments follow the same early stopping procedure.

| Changed parameter | Learning rate | Weight decay | L1 | Cutout | Early stopping |
|---|---|---|---|---|---|
| Learning rate | 1e-2 | 5e-4 | 0 | 0 | 120 |
| | 3e-2 | 5e-4 | 0 | 0 | 120 |
| | 1e-1 | 5e-4 | 0 | 0 | 120 |
| Weight decay | 1e-1 | 0e-4 | 0 | 0 | 120 |
| | 1e-1 | 1e-4 | 0 | 0 | 120 |
| | 1e-1 | 5e-4 | 0 | 0 | 120 |
| L1 | 1e-1 | 0 | 5e-7 | 0 | 120 |
| | 1e-1 | 0 | 2e-6 | 0 | 120 |
| | 1e-1 | 0 | 5e-6 | 0 | 120 |
| Cutout | 1e-1 | 0 | 0 | 4 | 120 |
| | 1e-1 | 0 | 0 | 8 | 120 |
| | 1e-1 | 0 | 0 | 12 | 120 |

Table F.2: Hyper-parameter settings in Figure F.2.

Figure F.3: **Ablation study on adversarially trained networks.** Re-implementing the measurements in Figure 3 for three other different choices of $\alpha$, $\beta$ and the attack size $\epsilon$. The parameters are provided in Table F.3. For all settings, robustness still increases with thickness.

## F.3 Different choices of the hyper-parameters in measuring adversarially trained networks

Here, we study the connection between boundary thickness and robustness under different choices of hyperparameters used to measure thickness. Specifically, we use three different sets of hyperparameters to reimplement the experiments that obtained Figure 3a and Figure 3b. These parameters are provided in Table F.3. The first row represents the base parameters used in Figure 3a and Figure 3b. Then, the second row changes $\beta$. The third and the fourth row change the attack size $\epsilon$ and step size. The changes in the hyper-parameters maintains our conclusion regarding the relationship between thickness and robustness. See Figure F.3.

| Figures | $\alpha$ | $\beta$ | Attack $\epsilon$ | Number of steps | Step size |
|---|---|---|---|---|---|
| Fig. 3a and Fig. 3b (the base setting in the main paper) | 0 | 0.75 | 2.0 | 20 | 0.2 |
| Fig. F.3a and F.3b | 0 | 0.5 | 2.0 | 20 | 0.2 |
| Fig. F.3c and F.3d | 0 | 0.75 | 1.0 | 20 | 0.1 |
| Fig. F.3e and F.3f | 0 | 0.75 | 0.6 | 20 | 0.06 |

Table F.3: Hyper-parameter settings in Figure F.3.

Figure F.4: **Boundary thickness versus margin.** Reimplementing procedures in Fig. 3c with different $\beta$ when $\alpha = 0$. There is a large range of $\beta$'s in which boundary thickness can measure robustness.

## F.4 Additional experiment on comparing boundary thickness and margin

Here, we further analyze how boundary thickness compares to the margin. In particular, we study what happens if we change the two hyper-parameters $\alpha$ and $\beta$ to the limit when boundary thickness becomes similar to margin, especially average margin shown in Figure 3c. The purpose of this additional experiment is to study the distinction between boundary thickness and average margin, even when they become similar to each other.

We notice that, when reporting boundary thickness, apart from the ablation study, we always use parameters $\alpha = 0$ and $\beta = 0.75$. When measuring average margin shown in Figure 3c, as we mentioned in Section 3.3, we approximately measure margin on the direction of adversarial examples, which effectively measures the boundary thickness when setting $\alpha = 0$ and $\beta = 1$. Thus, a reasonable questions is why the results of average margin in Figure 3c are fundamentally different from boundary thickness, despite the small change only in $\beta$.

To answer this question, we empirically show the transition from $\beta = 0.9$ to $1.0$. See Figure F.4. We see that a sudden phase-transition happens as $\beta$ gets close to $1.0$. This phenomenon has two implications. First, although boundary thickness reduces to margin with specific choices of parameters (in particular, $\alpha = 0$ and $\beta = 1$), it is different from margin when $\beta$ is not close to $1$. Second, for a large range of $\beta$, boundary thickness can distinguish robustness better than margin.

# G Addition Experiments and Details of Noisy Mixup

In this section, we first provide more details for the experiments on noisy mixup. Then, we provide some additional experiments to further compare noisy mixup and ordinary mixup.

## G.1 More details of noisy-mixup

In the experiments, we use ResNet-18 on both CIFAR-10 and CIFAR-100. For OOD, we use the datasets from [38] and evaluate on 15 different types of corruptions in CIFAR10-C and CIFAR100-C, including noise, blur, weather, and digital corruption.

The probability to replace a clean image with a noise image is 0.5. The model is trained for 200 epochs, and learning rate decay with a factor of 10 at epochs 100 and 150, respectively. For both mixup and noisy mixup, we train using three learning rates $0.1, 0.03, 0.01$ and report the best results. The weight decay is set to be 1e-4, which follows the recommendation in [16]. For noisy mixup, the value on each pixel in a noise image is sampled independently from a uniform distribution in [0,1], and processed using the same standard training image transforms applied on the ordinary image inputs.

| Dataset | Method | Clean | OOD | Black-box | PGD-20 | | |
|---|---|---|---|---|---|---|---|
| | | | | | 8-pixel | 6-pixel | 4-pixel |
| CIFAR10 | Mixup | **96.0**±0.1 | 78.5±0.4 | 46.3±1.4 | 2.0±0.1 | 3.2±0.1 | 6.3±0.1 |
| | Mixup-SEP | 95.3±0.2 | 82.1±0.9 | 55.5±8.6 | 4.4±1.6 | 6.5±1.9 | 11.3±2.4 |
| | Noisy mixup | 94.4±0.2 | **83.6**±0.3 | **78.0**±1.0 | **11.7**±3.3 | **16.2**±4.2 | **25.7**±5.0 |

Table G.1: **Mixup-SEP.** Having an additional "NONE" class in mixup-SEP can help improve robustness of mixup, but it still cannot achieve the robustness of noisy mixup. Results are reported for the best learning rate in [0.1, 0.03, 0.01].

| Dataset, Model | Method | Clean | OOD | Black-box | PGD-20 | | |
|---|---|---|---|---|---|---|---|
| | | | | | 8-pixel | 6-pixel | 4-pixel |
| CIFAR100, ResNet-18 | Mixup | **78.3**±0.8 | 51.3±0.4 | 37.3±1.1 | 0.0±0.0 | 0.0±0.0 | 0.1±0.0 |
| | Noisy mixup | 72.2±0.3 | **52.5**±0.7 | **60.1**±0.3 | **1.5**±0.2 | **2.6**±0.1 | **6.7**±0.9 |
| CIFAR100, ResNet-50 | Mixup | **79.3**±0.6 | 53.4±0.2 | 39.7±1.3 | 1.0±0.1 | 1.6±0.2 | 3.1±0.4 |
| | Noisy mixup | 75.5±0.5 | **55.5**±0.3 | **59.7**±1.1 | **4.3**±0.3 | **6.4**±0.1 | **10.3**±0.0 |

Table G.2: **Clean accuracy drop of noisy mixup on CIFAR100.** The drop of clean accuracy can be mitigated by using a larger ResNet-50 network. Results are reported for the best learning rate in [0.1, 0.03, 0.01].

When testing the robustness of the ordinary mixup and noisy mixup models, we used both black/white-box attacks and OOD samples. For white-box attack, we use an $\ell_\infty$ PGD attack with 20 steps. The attack size $\epsilon$ can take values in 8-pixel (0.031), 6-pixel (0.024) and 4-pixel (0.0157), respectively. The step size is 1/10 of the attack size $\epsilon$. For black-box attacks, we use ResNet-110 to generate the transfer attack. The other parameters are the same with the 8-pixel white-box attack.

When we measure the boundary thickness values of models trained using mixup and noisy mixup, we estimate each thickness value by repeated runs on 320 random samples. Then, we repeat this procedure 10 times and report both the average and three times the standard deviation in Figure 4.

## G.2 Addition experiments on noisy-mixup

Here, we provide two additional experiments on noisy mixup. In the first experiment, we want to study the effect of introducing an additional "NONE" class in noisy mixup. In the second experiment, we justify the clean accuracy drop of noisy mixup as a consequence of insufficient model size.

In the first experiment, we focus on the additional "NONE" class introduced in noisy mixup. We notice that we attribute the improved robustness to the mixing of the "NONE" class with the original images in the dataset. However, due to this additional class, it is possible that the network can have improved robustness by learning to distinguish clean images from noise. Thus, to separate the pure effect of having the additional "NONE" class in noisy mixup, we compare mixup and noisy mixup when mixup also has this additional class. Specifically, we measure ordinary mixup on CIFAR10 with the 11th class but only mix sample pairs within the first ten classes or within the 11th class. We call this method "mixup-SEP". See the results in Table G.1. Note that the results of mixup and noisy mixup are the same with Table 1. From the results, we see that noisy mixup is more robust than mixup-SEP. Thus, we cannot attribute the improved robustness solely to the "NONE" class.

In the second experiment, we further study the clean accuracy drop of noisy mixup shown in Table 1. As we have mentioned, one reason for the clean accuracy drop is the tradeoff between clean accuracy and robust accuracy often seen in robust training algorithms. The supporting evidence is that adversarial training in the same setting using ResNet-18 on CIFAR100 only achieve 57.6% clean accuracy. Here, we analyze another potential factor that leads to the drop of clean accuracy, which is the size of the network. To study this factor, we change ResNet-18 to ResNet-50 and repeat the experiment. We report the results in Table G.2. Note that the results for ResNet-18 are the same with Table 1. Thus, we can see that the drop of clean accuracy reduces when using the larger ResNet-50 compared to ResNet-18.

# H   More Details of the Experiment on Non-robust Features in Section 4.2

In this part, we provide more details of the experiment on non-robust features. First, we discuss the background on the discovery in [41]. Using a dataset $\mathcal{D}_{\text{train}} = \{(x_i, y_i)\}_{i=1}^n$, $\mathcal{D}_{\text{test}}$ and a $C$-class neural network classifier $f$, a new dataset $\mathcal{D}'$ is generated as follows:

- (**attack-and-relabel**) Generate an adversarial example $x_i'$ from each training sample $x_i$ such that the prediction of the neural network $y_i' = \arg\max_{j \in \{1,2,...,C\}} f(x_i')_j$ is not equal to $y_i$. Then, label the new sample $x_i'$ as $y_i'$. The target class $y_i'$ can either be a fixed value for each class $y_i$, or a random class that is different from $y_i$. In this paper, we use random target classes.

- (**test-on-clean-data**) Train a new classifier $f'$ on the new dataset $\mathcal{D}' = \{(x_i', y_i')\}_{i=1}^n$, evaluate on the original clean testset $\mathcal{D}_{\text{test}}$, and obtain a test accuracy ACC.

The observation in [41] is that by training on the completely mislabeled dataset $\mathcal{D}' = \{(x_i', y_i')\}_{i=1}^n$, the new classifier $f'$ still achieves a high ACC on $\mathcal{D}_{\text{test}}$. The explanation in [41] is that each adversarial example $x'$ contains "non-robust features" of the target label $y_i'$, which are useful for generalization, and ACC measures the reliance on these non-robust features. The test accuracy ACC obtained in this way is the *generalization accuracy* reported in Figure 5b.

In Figure 5b, the $x$-axis means different epochs in training a source model. Each error bar represents the variance of non-robust feature scores measured in 8 repeated runs. Thus, each point in this figure represents 8 runs of the same procedures of a non-robust feature experiment for a different source network, and each curve in Figure 5b contains multiple experiments using different source networks, instead of a single training-testing round. It is interesting to see that source networks trained for different number of epochs can achieve different non-robust feature scores, which suggests that when the decision boundary changes between epochs, the properties of the non-robust features also change.

In the experiments to generate Figure 5b, we use a ResNet56 model as the source network, and a ResNet20 model as the target network. These two ResNet models are standard for classification tasks on CIFAR10. The source network is trained for 500 epochs, with an initial learning rate of 0.1, weight decay 5e-4, and learning rate decay 0.1 respectively at epoch 150, 300, and 450. When training with a small learning rate, the initial learning rate is set as 0.003. When training with mixup, the weight decay is 1e-4, following the initial setup in [16]. The adversarial attack uses PGD with 100 iterations, an $\ell_2$ attack range of $\epsilon = 2.0$, and an attack stepsize of 0.4.

**Remark 7** (Why Thick Boundaries Reduce Non-robust Features)**.** Our explanation on why a thick boundary reduces non-robust feature score is that a thicker boundary is potentially more "complex"[4]. Then, in the attack-and-relabel step, the adversarial perturbations are generated in a relatively "random" way, independent of the true data distribution, making the "non-robust features" preserved by adversarial examples disappear. Studying the inner mechanism of the generation of non-robust features and the connection to boundary thickness is a meaningful future work.

## Footnotes

[4]Note that, although various complexity measures are associated with generalization in classical theory, and the inductive bias towards simplicity may explain generalization of neural networks [28, 29], it has been pointed out that simplicity may be at odds with robustness [27].