[Reviews · NeurIPS 2020]

Review 1

Summary and Contributions: The paper introduces the notion of "boundary thickness" for classifiers: a measure of the distance, in input space, between regions of confidence for classification decisions. This work shows how this measure can be harnessed for gaining insight into model robustness and its relation with decision boundaries. In particular, the paper conducts experiments that demonstrate (1) how a thicker decision boundary correlates with improved robustness against adversarial examples and out-of-distribution (OOD) examples, while the more fundamental notion of margin is unable to capture such properties, and (2) how conventional data augmentation and regularization schemes (and adversarial training) correlate with learning thicker decision boundaries. Further, the paper studies the relation between Mixup and boundary thickness, and proposes a noisy version of Mixup, which performs well against adversarial and OOD examples, and which is shown avoid learning non-robust features. - Update after discussion: several concerns raised by other reviewers are important, and some of them were addressed in the rebuttal. For the concerns, in particular, as R2 points out, a strong bias is induced when selecting x_s to be adversarial examples, which clearly limits the scope of the experimental results obtained. Regarding concerns raised by R3, I see that isolating of the effect of the noisy class in noisy mixup is interesting, and was indeed addressed in the rebuttal. With respect to the concerns pointed by R4, it is clear that there are several knobs to tune in the method, but also that the paper does a good job at studying these knobs. In summary, I consider the reviewers observed several issues that I was unable to see. These issues affect my view of the paper, hence I'll lower my score to 6, but still I consider that the concepts presented in the paper are valuable (and promising) enough to be published.

Strengths: * The proposal of another measure for quantifying properties of decision boundaries is of high interest to the whole ML community. Further, the study of robustness through the lens of this new measure is an adequate setup for this quantity. Such study allows to gain insight in both the phenomena and the measure. * Boundary thickness is theoretically well-founded, and its connection to the fundamental notion of margin is studied in this work. * Experiments show correlation between this quantity and robustness against adversarial and OOD examples, validating the usefulness of boundary thickness. * Experiments also show how common data augmentation and regularization schemes are connected to boundary thickness, which sheds light into why such schemes are useful for generalization and adversarial robustness. * The noisy version of Mixup proposed in this work shows promising results. * The relation between boundary thickness and how models focus on "non-robust features" is an interesting observation. * The Suppl. is a rather complete document with comprehensive experiments which provide further insight into boundary thickness.

Weaknesses: * It is unclear in the paper how measuring boundary thickness is "straightforward", first by itself and then compared to computing the margin. In my understanding, boundary thickness, in the case of general neural networks, is estimated by conducting adversarial attacks on various instances (Section E.1 in the Suppl.). Hence, it is an approximation, with rather few samples (320 samples are reported). This fact is important, and should be mentioned in the main paper, not only the Suppl. Further, is there a way of estimating how far this approximation is from the true value? * When generating the adversarial instance for the subsequent computation of boundary thickness, the paper mentions (L194) that the "size" of the attack is 0.1 (I understand this is the radius of the L2 ball around the original instance). Is this constraint setting an upper bound to the thickness? As no instance beyond this ball will be searched for. Would not this constraint affect the subsequent estimation of the overall boundary thickness? The following are rather minor comments * Computing boundary thickness requires defining a distribution from which pairs of instances will be sampled. The paper proposes (and studies) (1) sampling one instance randomly and computing an adversarial example from it, and (2) sampling two random instances with different labels (Section E.2 in the Suppl.). While both distributions are of interest, it appears that there is much left to be studied, as the "adversarial direction" is most likely sub-optimal for computing boundary thickness.

Correctness: I see no problem with the methodology or the conclusions that are drawn from experiments (besides extra experiments that come to mind for exploring other dimensions of/with the boundary thickness notion). However, I am inclined towards discussing perhaps some dimension other reviewer noticed and I missed.

Clarity: Yes, the paper is well written and easy to follow (despite the large amount of theoretical and experimental findings). The paper appears to be the result of extensive experiments and theoretical work, and so the Suppl. is fundamental.

Relation to Prior Work: Yes, appropriate references are given for understanding the context in which this work is proposed.

Reproducibility: Yes

Additional Feedback: * A pairwise-class extension of boundary thickness could be of interest. Perhaps it could help diagnose problems in robustness arising from boundaries between particular classes. * Contemporaneous work, "Adversarial Vertex Mixup: Toward Better Adversarially Robust Generalization: CVPR2020, Lee et al. could be of interest Minor issues: L48: can convey, at first, that noisy mixup is not a contribution of this work L85: perhaps "convex" is more precise L85: in the original paper the limits were inclusive Figure 1d: in Figs (b) and (c) the thickness was always larger than the margin, in (d) this is reversed. Could be confusing. L110: gap or length? Eq (6): it is my understanding that T(u) reaches a maximum when there is *no* tilting. Is this correct? T(u) quantifies the opposite of tilting? This could be confusing Figure 2(a): axis labels are hard to read. "DenseNet" does not have correct type-setting. Reporting evolution of test set accuracy could also be informative Figure 2(b): from what epoch are these models? How were they chosen? What is their clean test set accuracy? Figure 3: dataset (and model) are not stated in the caption. L232: Intensity of the color is related to thickness or robustness? I had understood it was robustness, but the text states it was thickness. L235 (whole paragraph): Explanation regarding the computation of margin could be more useful before referring the reader to the figure. L259: unclear what type of noise is used for noisy mixup (distribution and parameters). L267 (and Table 1): 8/4/2-pixel incites to think of number of pixels (like L0 norm) rather than 8/4/2 intensities (like L_\infty norm), which is what I understand is actually being used in this experiment. L293: The sentence could make clearer what "this sense" means. L297: unclear what the original (larger) learning rate was. Few grammar-related issues: L55: end with "." instead of ":" L66: makes L131-132: "The thicker boundary [...] is harder to fit data" L138: "SVM" abbreviation is used before introducing complete form Eq (3): "Margin" typesetting is different from that in Eq (4) L179: Could be clearer as to what "training algorithm" is being referenced L185: differentiate different (cacophonic) L188-190: are CIFARs, ResNets, DenseNets and VGGs missing from references? L207: is SVHN missing from references? Table 1: CIFAR was spelled "Cifar" L244, Figure 5 (caption), L280: phenomena L262: combintions References: several works appear as their arXiv version. Probably were also published in a venue.


Review 2

Summary and Contributions: Authors present a new concept of decision boundary thickness to characterize generalization and robustness of a classifier. Decision boundary is defined as a distance between level sets of the posterior probability difference function (p(0|x) - p(1|x)). Authors show that boundary thickness is equal to margin in the case of linear SVM and reduces boundary tilting (i.e overfitting to data manifolds). Authors also conduct comprehensive empirical validation of their claim. They show direct relationship between classifier robustness to increasing boundary thickness. They also demonstrate that boundary thickness can distinguish between different levels of adversarial perturbations while the margin metric cannot. Lastly, they present a new training scheme, noisy mixup, which augments the data by a convex combination with a noisy image. These scheme shows robustness to adversarial attacks when compared to regular mixup. Authors also explore connection to robust features as an explanation to adversarial examples.

Strengths: An interesting paper. The concept of boundary thickness is novel and appears to do a better job of measuring robustness (and more straightforward to compute) than margin. This is shown experimentally. A noisy mix up training scheme also looks promising. Robustness and adversarial perturbation is an important aspect of modern ML so this is a worthwhile area to investigate.

Weaknesses: It would be good to incorporate the concept as a modification to the training objective (perhaps in addition to the noisy mixup data augmentation scheme). The last section discussing connection to robust features / explaining robust phenomena could be clarified. I don't really get figure 5 (b). Is it saying mixup does not improve generalization accuracy?.

Correctness: Looks correct

Clarity: clear enough

Relation to Prior Work: related work is described.

Reproducibility: Yes

Additional Feedback: I reviewed authors' feedback and will be keeping my current score.


Review 3

Summary and Contributions: This paper proposes boundary thickness as a metric for assessing classifiers. Boundary thickness is a generalization of margin and measures the thickness of decision boundaries between two sets of data points. The paper introduces theoretical justifications for this metric and can establish some claims on linear classifiers. The paper shows empirically that boundary thickness correlates with adversarial and OOD robustness and shows that common regularization and data augmentation methods result in thicker boundaries. Lastly, the paper introduces noisy mixup, a novel augmentation scheme that improves boundary thickness and leads to more robust models.

Strengths: The paper grounds their claims well in theory. In fact, the paper spends a lot of effort convincing the reader that in linear models, where theoretical claims can be made, boundary thickness is a sensible measure and leads to optimally robust classifiers. For example, it optimally mitigates boundary tilting in a linear binary classifier. The empirical evaluation is also sound and all claims that are made are supported by corresponding evidence. The work is significant and relevant and novel, because adversarial robustness is a core topic and both the ability to measure it and to improve it are valuable contributions.

Weaknesses: Please consider my score as "undecided" rather than a fixed evaluation. I'm very open to being convinced by you or the other reviewers and change my mind. - What I don't like about the empirical evaluations is that clean (unperturbed test set) accuracy is reported and analyzed very sparingly. It is sometimes alluded to in the text, but in analyzed systematically, which is something that is extremely important when comparing the adversarial robustness of different methods. Effects can arise or vanish purely because of the fact that a model was regularized more / less by any given method and has given up less / more of its clean accuracy. I understand that you state in the text that you've measured clean accuracy and the effect you're observing isn't explained by that, but I would still like to see a much more thorough analysis of it throughout the paper. For example, in Table 1 it would be sensible to have an additional column to report clean test accuracy. - You state that "margin is a special case of boundary thickness", which is not really the case as I see it. Yes, the quantity that is measured can be rephrased to match, but boundary thickness takes the expectation over a distribution, whereas margin takes the minimum, which is actually quite a significant difference. So maybe you can say they are related, or say that they are the same in some limit case, but as such, they aren't. Please correct me if I'm wrong. - Figure 1d makes the distinction between boundary thickness and margin clear, and I like that. However, for you experiments you then use alpha = 0, which makes the two a lot more close to each other as now the only difference is beta (0.75 for you, 1 for margin) and the minimum / expectation difference. If you do this universally, it might be advantageous to adjust the figure to reflect that. Also, how exactly did you measure the average margin in figure 3c? Because what I would have suggested is that you replace the minimum condition in the margin with the same expectation you have in boundary thickness and it seems like you've done this here, but then it seems very strange that there is such a big difference in results, given that the only difference now is beta=0.75 vs beta=1. This leads me to believe that the average margin is something else and the text is very unclear on this. - It seems like boundary thickness isn't so much a measure of actual geometric width, rather than a measure of the slope of the decision function between two places. Am I correct to assume that if you had a classifier where decision boundaries are very steep (i.e. step-functions), but nevertheless classes are separated very well, boundary thickness would tell me that the boundary is very thin? In addition, margin would correctly tell me that margin is wide in this case? Also the model would be quite robust in this case? It seems like you're relying on the fact that the training procedure makes these decision boundaries less steep, the farther the data is apart. Is that a valid assessment? - In your proposed noisy mixup method, you add the additional noise class, which I can see leads to good results. But also, now you have two different effects: one of mixup with this additional class and one by the pure fact that you have this additional class (and therefore some inherent OOD detection properties). You could isolate this by also giving mixup the additional class, but then simply regulate up and down by how much you mix with this additional class. - As for your theoretical justification for noisy mixup, you justify mixup very well, but the addition of the noise class, which is the core of your method, is introduced in a hand-wavey way. It seems a bit arbitrary and I don't see a direct reason why this particular choice is the most obvious one to make the boundary thicker. What would be more supporting of your general paper is a new method that explicitly tries to increase the thickness metric during training and then show that that has some of the claimed properties. - Minor: In line 252 you claim you show a minimax bound, but then you show a maxmin bound. - The first section in 4.2 seems a bit out of place. It is too short for me to accurately understand what's going on. I realize space is tight and you want to show as much results as possible. Maybe try to rephrase. For example, I have no clue what "boundary thickness measured on saturated images in adversarial training" is supposed to mean. - Boundary thickness is motivated well theoretically, but in practice you need to choose your x_s, which you always choose as l2-adversarial examples. I would expect the paper to go more into the choice of this and what effects and potential shortcomings it has, though I recognize that you only have 8 pages, so that's maybe an unreasonable requirement. Minor: In L_inf attacks, the eps-limits are not "pixels", (i.e. you say "eps = 8 pixels"), but units in the 256-bit color space.

Correctness: The claims are correct. With respect to the empirical methodology, my only concern is the lack of reporting of clean accuracy, which I believe to be crucial.

Clarity: The paper is well written, but sometimes tries to tell too much in little text, especially towards the end, such that it gets unclear. This is most likely a result of the page limit.

Relation to Prior Work: It introduces a new metric and a new augmentation method.

Reproducibility: Yes

Additional Feedback: POST-FEEDBACK UPDATE: Thank you for your feedback, especially for the clarifications and additional experiments. I'm genuinely convinced that you've found something that correlates with OOD and adversarial robustness in your setups. I've updated my score leaning towards accept. The reason it's not higher is that I am not convinced that this connection can be seen as causal or is due to the intuitions you give, for that you'd need to develop a method that directly increases BT and then show it results in more robustness (or develop a training method that tries to be high BT while being non-robust and then show that that method always fails).


Review 4

Summary and Contributions: This paper introduces the notion of the boundary thickness of a classifier, which generalizes the notion of margin. It is claimed that thick decision boundaries lead to improved accuracy and higher robustness against adversarial examples and out-of-distribution transforms.

Strengths: The notion of boundary thickness is novel to the NeurIPS community for measuring the expected distance to travel along line segments between different classes across a decision boundary. This paper shows that some regularization and data augmentation methods can increase the boundary thickness. Also it indicates that maximizing boundary thickness is akin to minimizing the so-called mixup loss, and the authors proposed a noise augmentation method on mixup training to increase boundary thickness.

Weaknesses: There are several serious weaknesses from my point of views. 1. The definition of boundary thickness is rather complex and might hinder the understanding of this notion and the practical use. How to choose \alpha and \beta? How to select the distribution p? It might happen that boundary thickness of classifier A is better than that of classifier B under one set of parameters, but changing another set of parameters this case will not hold. 2. I think the theory about boudary thickness is lacking in this paper right now. How about the generalization bound with respect to boundary thickness? Is there anything like the generalization bound of margin? 3. Can we develop new algorithms that aim to maximize the boundary thickness? I did not see any specific algorithm from this motivation. 4. The practical importance of this new notion has not been demonstrated in the empirical experiments. The authors devised the noisy-mixup training method and showed the improved robustness. But I think there is a weak connection with this new notion of boundary thickness, at most the authors showed boundary thickness has been enlarged empirically and the improved robustness can somehow attribute to the enlarged thickness.

Correctness: I did not fully check the proofs of the three propositions in this paper. It seems that the claims and method correct. But this is not the emphasis. The above four weaknesses hurt the quality of this paper.

Clarity: The paper is well written, but some important aspects are lacking from my point of view.

Relation to Prior Work: I am satisfied with the presentation about previous work. However the authors may reference the following paper that discusses the margin, slope, and curvature of the decision boundary: Supervised Learning via Euler's Elastica Models, JMLR 2015.

Reproducibility: Yes

Additional Feedback:

[Author Response · NeurIPS 2020]

Two reviewers had high scores. We will not focus on their minor comments here but will incorporate their suggestions.
One reviewer was undecided and had an intermediate score due to detailed questions we will address. The lowest score
was given by a reviewer who had rather general concerns. These are all important for follow-up work, as we describe.

**R1:** Is computing boundary thickness (BT) straightforward? A: Estimating BT (Eq.(1)) on the adversarial direction
does not need the exact projection required for margin (Eq.(3)), so it is straightforward, compared to measuring the
exact margin. One can obtain the Std Dev of BT estimate using repeated runs. E.g., Fig 4 uses 10 runs and each uses
320 samples. We tried increasing from 320 samples to 1280, which reduces the Std Dev further from 0.037 to 0.020.

**R2:** Fig 5b. A: The y-axis is the generalization using non-robust features only, as done by [33] (details in Suppl Sec H).

**R3:** Clean accuracy (acc). A: Clean acc is critical, and we will report it in the main text. However, the robustness of
noisy mixup does not arise purely due to increased regularization. To demonstrate, we vary the weight decay in mixup to
obtain different levels of regularization and report clean versus robust acc in Figure-R 1 (1). Noisy mixup outperforms
mixup with varying regularization levels. Table 1 clean acc: (CIFAR10) mixup/noisy mixup is 96.0/94.4, (CIFAR100)
mixup/noisy mixup is 78.3/72.2. The drop of clean acc is expected for robust models [Dan, Wei, Ravikumar, 2020].
E.g., we tried the same network (ResNet18) with adversarial training and achieved 57.6 clean acc on CIFAR100, which
is about 20% drop. Another reason for the clean acc drop is the network size, because when we change ResNet18 to
ResNet50, the clean acc drop reduces. Specifically, for mixup/noisy mixup using ResNet50, the clean acc is 79.3/75.5,
the OOD acc is 53.4/55.5, the robust acc under PGD-20 is (1.0, 1.6, 3.1)/(4.3, 6.4, 10.3) with (8, 6, 4)-unit attack.

**R3:** Margin takes the minimum and boundary thickness (BT) takes average. A: The classical margin does take minimum,
but average margin has also been considered. e.g., see Eq.(9) in [13]. When using their "sum aggregation", [13] averages
over both samples and classes. Prop 2.2 stresses that it is the "min version" of BT that reduces to classical margin. We
empirically compare BT to both average and minimum margin, see Fig 3c.

**R3:** Average margin measurement in Fig3c, and large change in BT from $\beta = 0.75$ to $\beta = 1$. A: The average margin
in Fig 3c is indeed measured as the expectation. Figure-R 1 (2-6) shows the transition from $\beta = 0.9$ to 1.0. A huge and
sudden phase-transition happens as $\beta$ gets 1.0. This is why BT as we define performs better than margin.

**R3:** Geometric interpretation of BT. Steep boundary. A: Geometrically, BT refers to weighted average distance
between pairs $(x_r, x_s) \sim p$, with weights defined by how much mass of posterior probability difference is captured
between level sets $[\alpha, \beta]$. This implies the "slope" of decision function is somewhat related to BT, as pointed by R3. It
is hard to solely attribute this to the slope because high dimensional optimization landscape on non-separable data can
be tricky. Furthermore, the slope at the decision boundary is a local property, and the BT is more of a global metric
that we agree is at least somewhat related to the former. In the toy case when the boundary is steep and classes are
well-separated, margin (i.e. BT with $\alpha = 0, \beta = 1$) captures robustness well while $\alpha = 0, \beta = 0.75$ leads to thinner BT
as the reviewer points out. But this motivation for margin is too simplistic and is not representative of optimization
landscapes in practical problems. E.g., we observe that NNs with a steep boundary overfit and have low robustness,
even in adversarial training (see the left-most points in Fig 3a that have small BT and low robustness). Our contribution
is to capture robustness for practical NNs, and we provide two knobs ($\alpha$ and $\beta$) to generalize the simplistic concept
of margin in a principled way. We observe through exhaustive empirical evidence and ablation studies that setting
$\alpha = 0, \beta = 0.75$ is a much better indicator for robustness than margin (when $\alpha = 0, \beta = 1$).

**R3:** Isolating effect of introducing the noise class. A: To separate the pure effect of the additional class in noisy
mixup, we follow R3's advice and measure ordinary mixup on CIFAR10 with the 11th class but only mix sample pairs
within the first ten classes or within the noise class. The clean acc of this scheme is 95.3%, the OOD acc is 82.1%, the
black-box robust acc is 55.5%, and the robust acc under PGD-20 attack with (8, 6, 4) units is (4.4, 6.5, 11.3). Comparing
with Table 1, we see that noisy mixup is more robust in all of the aspects above.

**R4:** Boundary thickness (BT) definition is complicated; choices of hyper-parameters. Generalization theory. A: There
is subtlety to our definition, but we have tried to explain the intuition and relationship of BT with other concepts, while
being mathematically precise. See discussions with other reviewers and their reviews on clarity of the paper. Regarding
choosing hyperparameters, we have provided extensive ablation study on choosing different hyper-parameters, including
$\alpha, \beta$ (Section E.3.2 and Section F.3), and $p$ (Section E.2). In all experiments, a thick boundary improves robustness.
Generalization theory is important follow-up, but it is outside the scope of this paper. We hope this reviewer can
conclude that our (empirical and theoretical) findings are worthy of being shared with the ML community.

**R4:** Other algorithms to explicitly increase BT. Noisy mixup is weakly connected. A: We have justified theoretically
for using mixup to increase BT during training (Prop. A.1). Noisy mixup indeed explicitly increases BT because it
increases BT both amongst training samples (like mixup) as well as between training samples and noise samples. We
agree that there may be other better ways to increase BT for OOD errors than just using the mixup on noise class.
However, we show that this simple scheme works reasonably well, and also show the corresponding increase in BT in
Fig 4. We hope that by introducing BT to the community, our work motivates further work in this direction.

Figure-R 1: For R3: (1) Comparing noisy mixup and mixup with different weight decay. Varying the weight decay ($\ell_2$ regularization) only cannot match the performance of noisy mixup. (2-6) Reimplementing Fig 3c with different $\beta$ when $\alpha = 0$. There is a large range of $\beta$'s in which boundary thickness can measure robustness.

[Meta-Review · NeurIPS 2020]

This proposal relies on the concept of boundary thickness to explain robustness and provides a convincing experimental study. The reviewing committee recommends the acceptance of this paper. We suggest the authors incorporate the additional points included in the rebuttal to the final version of the paper. In particular, given the proximity with the concept of margin and the vast literature on the topic, we recommend the authors clarify the differences as they did in the rebuttal.